# Towards the Fundamental Limits of Knowledge Transfer over Finite Domains

**Qingyue Zhao**
Department of Computer Science and Technology
Tsinghua University
`zhaotsingyue@gmail.com`

**Banghua Zhu**
Department of Electrical Engineering and Computer Sciences
University of California, Berkeley
`banghua@berkeley.edu`

## Abstract

We characterize the statistical efficiency of knowledge transfer through $n$ samples from a teacher to a probabilistic student classifier with input space $\mathcal{S}$ over labels $\mathcal{A}$. We show that privileged information at three progressive levels accelerates the transfer. At the first level, only samples with hard labels are known, via which the maximum likelihood estimator attains the minimax rate $\sqrt{|\mathcal{S}||\mathcal{A}|/n}$. The second level has the teacher probabilities of sampled labels available in addition, which turns out to boost the convergence rate lower bound to $|\mathcal{S}||\mathcal{A}|/n$. However, under this second data acquisition protocol, minimizing a naive adaptation of the cross-entropy loss results in an asymptotically biased student. We overcome this limitation and achieve the fundamental limit by using a novel empirical variant of the squared error logit loss. The third level further equips the student with the soft labels (complete logits) on $\mathcal{A}$ given every sampled input, thereby provably enables the student to enjoy a rate $|\mathcal{S}|/n$ free of $|\mathcal{A}|$. We find any Kullback-Leibler divergence minimizer to be optimal in the last case. Numerical simulations distinguish the four learners and corroborate our theory.

## 1 Introduction

It has become common sense that transferring intrinsic information from teachers to the greatest extent can expedite a student's learning progress, especially in machine learning given versatile and powerful teacher models. Learning with their assistance has been coined *knowledge distillation* (KD) (Hinton et al., 2015; Lopez-Paz et al., 2015), a famous paradigm of knowledge transfer leading to remarkable empirical effectiveness in *classification* tasks across various downstream applications (Gou et al., 2021; Wang & Yoon, 2021; Gu et al., 2023b). The term *distillation* implies a belief that the inscrutable teacher(s) may possess useful yet complicated structural information, which we should be able to compress and inject into a compact one, i.e., the student model (Breiman & Shang, 1996; Buciluǎ et al., 2006; Li et al., 2014; Ba & Caruana, 2014; Allen-Zhu & Li, 2020). This has guided the community towards a line of knowledge transfer methods featuring the awareness of teacher training details or snapshots, such as the original training set, the intermediate activations, the last-layer logits (for a probabilistic classifier), the first- or second-order derivative or statistical information, and even task-specific knowledge (Hinton et al., 2015; Furlanello et al., 2018; Cho & Hariharan, 2019; Zhao et al., 2022; Romero et al., 2014; Zagoruyko & Komodakis, 2016; Yim et al., 2017; Huang & Wang, 2017; Park et al., 2019; Tian et al., 2019; Tung & Mori, 2019; Qiu et al., 2022; Srinivas & Fleuret, 2018; Cheng et al., 2020; Liang et al., 2023).

However, in more general modes of knowledge transfer, the procedure- or architecture-specific information can be unavailable, irrelevant, or ill-defined. Modern proprietary large language models (LLMs) like GPT-4 (OpenAI, 2023b) and Claude 2 (Anthropic, 2023) strictly confine the data returned through their APIs except the generated tokens, not to mention their training sets. Therefore, there has been an effort towards distilling LLMs only through oracle queries via locally crafted prompts (Zheng et al., 2023; Peng et al., 2023). In denoising (Tsybakov, 2003), boosting (Freund & Schapire, 1995), and prediction with expert advice (Cesa-Bianchi et al., 1997), the teacher group itself is the classes of interest and the student only gets positive-valued feedback from the teacher group, so the teacher implementation is immaterial. In certain circumstances of robot learning (Abbeel & Ng, 2004; Ross & Bagnell, 2010), the teacher demonstration may come from threshold-based classifiers or even human experts, where no teacher probability is defined.

To unify different natures of teacher information acquisition in knowledge transfer and assess the technical barriers in a principled way, we decompose the *transfer set*[1] into a generative model $\pi^\star(\cdot|\cdot)$ giving one $a$ in the label space $\mathcal{A}$ given a query $s$ in the input space $\mathcal{S}$ and the additional information provided by the teacher for each sample pair $(s,a)$.[2] Take several LLMs as examples. In OpenAI's chat completion API[3] for GPT-4, only responses are returned given a prompt $s$, in which no privileged information is accessible. An early completion API[4] for GPT-3 (Brown et al., 2020), however, also returns the log-likelihood of every token in the generated response. If with open-sourced models like GPT-2 (Radford et al., 2019) in hand, practitioners can extract the last-layer logits of each position in the sampled sequence as the privileged information. These typical examples reveal a trend that *the more powerful a foundation model (Bommasani et al., 2021) is, the less likely it is to release privileged information as a teacher model*[5], which naturally motivates a question:

*What is the fundamental limit of knowledge transfer given limited privileged information in general?*

In this work, we answer this question with minimal inductive bias by analyzing three classic cases with easier difficulties in order over finite $\mathcal{S}$ and $\mathcal{A}$. Concretely, for any learner $\widehat{\pi}$, we study the total variation between it and the reference policy $\pi^\star$ conditioned on an input distribution $\rho$, i.e.,

$$\sum_{s\in\mathcal{S}}\rho(s)\mathsf{TV}(\widehat{\pi}(\cdot|s),\pi^\star(\cdot|s))=:\mathsf{TV}(\widehat{\pi},\pi^\star|\rho),$$

where $\widehat{\pi}$ can access $n$ samples $\{(s_i,a_i)\}_{i=1}^n$ and optionally certain fraction of $\pi^\star(\cdot|s_i),\forall i\in[n]$.

## 1.1 MAIN CONTRIBUTIONS & CLARIFICATIONS

Table 1 gives an overview of rate bounds in distinct data acquisition protocols. Hard Labels indicates $\pi^\star$ to be a black-box for $\widehat{\pi}$ throughout the learning process. Partial SLs stands for *partial soft labels*, which means the teacher provides the student with an extra (partial) ground truth $\pi^\star(a|s)$ for every sample $(s,a)$. Soft Labels is a synonym of the logits on the entire $\mathcal{A}$ exposed to $\widehat{\pi}$ given each $s$ in $\{s_i\}_{i=1}^n$. Intuitively, each of the three levels discloses richer information than previous ones, which at least cannot make it harder to learn. Rigorously, progressively faster minimax rates are matching exactly at all levels[6] and every high-probability upper bound has its upper confidence radius at most the same order of the corresponding expectation lower bound up to polylog factors.

Table 1: The second column shows one out of $n$ data points available to the student. Worst-case bounds in the last column hold with high probability and $\widetilde{O}(\cdot)$ hides polylog factors.

| Transfer via | Data Format | Lower Bound | Loss | Performance |
|---|---|---|---|---|
| Hard Labels | $(s,a)$ | $\Omega\left(\sqrt{|\mathcal{S}||\mathcal{A}|/n}\right)$ 
 Theorem 3.1 | $\mathsf{CE}_{\mathsf{sgl}}$ | $\widetilde{O}\left(\sqrt{|\mathcal{S}||\mathcal{A}|/n}\right)$ 
 Theorem 3.2 |
| Partial SLs | $(s,a,\pi^\star(a|s))$ | $\Omega(|\mathcal{S}||\mathcal{A}|/n)$ 
 Theorem 4.1 | $\mathsf{CE}_{\mathsf{ptl}}$ 

 $\mathsf{SEL}_{\mathsf{ptl}}$ | $\Omega(1)$ 
 Theorem 4.3 
 $\widetilde{O}(|\mathcal{S}||\mathcal{A}|/n)$ 
 Theorem 4.4 |
| Soft Labels | $(s,a,\pi^\star(\cdot|s))$ | $\Omega(|\mathcal{S}|/n)$ 
 Theorem 5.1 | $\mathsf{CE}_{\mathsf{ful}}$ | $\widetilde{O}(|\mathcal{S}|/n)$ 
 Theorem 5.2 |

Technically, the later two protocols are nonstandard in density estimation, especially in terms of the derivation of minimax lower bounds in that we assume the data generating process to be $(\rho\times\pi^\star)^n$. The

---

[1]Here we mean the information a student has access to in total.

[2]Inspired by the *learning using privileged information* framework (Vapnik et al., 2015; Lopez-Paz et al., 2015).

[3]https://platform.openai.com/docs/api-reference/chat

[4]https://platform.openai.com/docs/api-reference/completions

[5]See, e.g., Ramesh et al. (2021; 2022); OpenAI (2023a) in computer vision for a similar tendency.

[6]We defer the matching expectation bounds to respective sections.

constructive proof (Appendix C.3) of Theorem 4.1 may be of independent interest. The performances at the second level are also more tricky. Theorem 4.3 sentences a naive adaptation of the cross-entropy loss ($\mathsf{CE}_{\mathsf{ptl}}$) to be a misfit with probability 1 when the privileged information is at a modest level.

We do not require $\pi^\star$ to be trained in any sense and assume $\widehat{\pi}$ to have no awareness of the teacher implementation, inspired by the aforementioned practical trend. Consequently, we do no distinguish the *teacher probability* from the *Bayes probability* (Menon et al., 2020; Dao et al., 2021) and $\rho$ can have no relation with teacher training. We idealize the samples to be *i.i.d.*, while all the results already extend to the setting where different $(s,a)$ pairs are mildly dependent, of which we refer the readers to Appendix G for detailed discussions.

We give more discussions on the connections between our framework and previous formulations in Appendix A.2 and review key related works aiming at the principles of knowledge transfer in the next subsection.

## 1.2 RELATED WORKS ON UNDERSTANDING KNOWLEDGE TRANSFER

Hinton et al. (2015) refers to the logits generated on a dataset by a teacher model, who has been trained using the same dataset, as soft labels; and refers to $\{a_i\}_{i=1}^n$ in this dataset as hard labels. Our terms, however, pay no attention to whether $\rho$ matches the original dataset; and our Soft Labels strictly carry more information (of $\rho \times \pi^\star$) than our Hard Labels, both of whose inputs $\{s_i\}_{i=1}^n$ are sampled from $\rho$. So the third column of Table 1 does not account for the *class similarities* argument (Furlanello et al., 2018), the *regularization effect via label smoothing (LS)* argument (Yuan et al., 2020; Tang et al., 2020), or the *bias-variance trade-off* argument (Zhou et al., 2021; Menon et al., 2020); all of which are classical wisdom explaining the benefit of soft labels in KD (Hinton et al., 2015).

These previous views are neither consistent nor complete for justifying why soft labels improves student training. (Müller et al., 2020) undermines the hypothesis that class similarities in soft labels are vital by the effectiveness of KD in binary classification. Han et al. (2023) challenges the regularization via LS thesis through better interpretablity-lifting effect of KD than LS. Dao et al. (2021) develops the bias-variance trade-off perspective in more complex scenarios. In contrast, we define Soft Labels (similarly Partial SLs) and tackle its boost over Hard Labels rigorously following the information-theoretic direction of the data processing inequality (Polyanskiy & Wu, 2022). We survey more works unraveling knowledge transfer. Other empirical and enlightening paradigms are deferred to Appendix A.

Most analyses of KD with trained deep nets as the teacher (Phuong & Lampert, 2019; Ji & Zhu, 2020; Panahi et al., 2022; Harutyunyan et al., 2023) lie in the linear or kernel regime, notably except Hsu et al. (2021), which finds the student network to have fundamentally tighter generalization bound than the teacher network under several nonlinear function approximation schemes. There are also works analyzing knowledge transfer between neural networks of the same architecture (Mobahi et al., 2020; Allen-Zhu & Li, 2020). Our framework goes exactly in the reverse direction: our analysis is not restricted to parsimonious students, over-parameterized teachers; or the *compression* subconcept of knowledge transfer (Buciluǎ et al., 2006; Bu et al., 2020). For example, a human demonstrator, who does not learn only from data and is able to output probabilistic belief, can also fit into the first column of Table 1 as a kind of teacher under our specification of $\pi^\star$.

## 2 PRELIMINARIES

**Notation.** For two nonnegative sequences $\{a_n\}$ and $\{b_n\}$, we write $a_n = O(b_n)$ or $a_n \lesssim b_n$ if $\limsup a_n/b_n < \infty$; equivalently, $b_n = \Omega(a_n)$ or $b_n \gtrsim a_n$. We write $c_n = \Theta(d_n)$ if $c_n = O(d_n)$ and $c_n = \Omega(d_n)$. For a metric space $(\mathcal{M}, d)$, $\mathsf{dist}_d(\cdot, \mathcal{N}) := \inf_{y \in \mathcal{N}} d(\cdot, y)$ for any $\mathcal{N} \subset \mathcal{M}$. The term *alphabet* is a synonym of *finite set*. For a set or multiset $\mathcal{C}$, let $|\mathcal{C}|$ denote its cardinality. For any alphabet $\mathcal{X}$, on which given two distributions $p$, $q$, the *total variation* between them is $\mathsf{TV}(p, q) := 0.5 \|p - q\|_1$, their *Kullback-Leibler (KL) divergence* is $\mathsf{KL}(p\|q) := \mathbb{E}_p[\log p - \log q]$; and we denote all $|\mathcal{X}|$ Dirac distributions on $\mathcal{X}$ by $\mathsf{Dirac}(\mathcal{X})$, where $\mathsf{Dirac}(\mathcal{X}, x)$ stands for the one concentrated at $x$. For two alphabets $\mathcal{X}$ and $\mathcal{Y}$, we denote by $\Delta(\mathcal{Y})$ the probability simplex on $\mathcal{Y}$ and define $\Delta(\mathcal{Y}|\mathcal{X}) := \{\pi(\cdot|\cdot) : \mathcal{X} \to \Delta(\mathcal{Y})\} \Rightarrow \pi(\cdot|x) \in \Delta(\mathcal{Y}), \forall x \in \mathcal{X}$.

## 2.1 COMMON SETUP

The teacher always exposes to the student a multiset $\mathcal{D} = \{(s_i, a_i) \in \mathcal{S} \times \mathcal{A}\}_{i=1}^n$ consisting of $n$ *i.i.d.* input-label tuples. To analyze $\mathcal{D}$ in a fine-grained way we introduce for $\mathcal{X}^n \ni X^n \overset{i.i.d.}{\sim} \nu$ the *number of occurrences* $\mathfrak{n}_x(X^n)$ of $x$ and the *missing mass* $\mathfrak{m}_0(\nu, X^n)$, which measures the portion of $\mathcal{X}$ never observed in $X^n$ (McAllester & Ortiz, 2003). We refer to the input (resp. label) component $\{s_i\}_{i=1}^n$ (resp. $\{a_i\}_{i=1}^n$) of $\mathcal{D}$ as $\mathcal{S}(\mathcal{D})$ (resp. $\mathcal{A}(\mathcal{D})$) and also define a multiset $\mathcal{A}(\mathcal{D}, s)$ for every $s \in \mathcal{S}$ to denote the $a_i$'s in $\mathcal{D}$ that are associated with the visitations of $s$, taking into account multiplicity.[7]

## 2.2 QUANTITY OF INTEREST

We assume $\mathcal{D} \sim (\rho \times \pi^\star)^n$ follows a product measure, where $\pi^\star \in \Delta(\mathcal{A}|\mathcal{S})$ is the ground truth distribution over $\mathcal{A}$ given $s \in \mathcal{S}$ the teacher holds and $\rho$ is some underlying input generating distribution. No assumption is imposed on the data generating process $\rho \times \pi^\star$ except for belonging to

$$\mathcal{P} := \{\rho \times \pi^\star : \rho \in \Delta(\mathcal{S}), \pi^\star \in \Delta(\mathcal{A}|\mathcal{S})\}.$$

In this work, we evaluate the performance of a student $\widehat{\pi}$ based on the TV between $\widehat{\pi}$ and $\pi^\star$ conditioned on $\rho$ defined as

$$\mathsf{TV}(\widehat{\pi}, \pi^\star | \rho) := \mathbb{E}_{s \sim \rho}[\mathsf{TV}(\widehat{\pi}, \pi^\star)], \tag{2.1}$$

though the student is never allowed to access $\rho$ directly. We investigate the convergence rate of (2.1) among three categories of students told by the teacher's degree of openness in the tabular setting. Intuitively speaking, all the learning procedures of interest try to match the log-probability kernel $\log \pi$, a notion of normalized *logits*, between the student and the teacher, especially via variants of the cross-entropy loss, which is standard in the study of classification both theoretically and practically (Paszke et al., 2019). Besides the universal definition $\mathsf{CE}_{\mathsf{ful}}(p\|q) := -\mathbb{E}_p[\log q]$ of the cross-entropy between $p \ll q$, a popular counterpart for hard labels specialized to classifiers is commonly defined as $\mathsf{CE}_{\mathsf{sgl}}(s, a; \pi) := \mathsf{CE}_{\mathsf{ful}}(\mathrm{Dirac}(\mathcal{A}, a)\|\pi(\cdot|s)) = -\log \pi(a|s)$.

## 3 TRANSFER VIA HARD LABELS

This is equivalent to the standard setting for estimating the conditional density $\pi^\star(\cdot|\cdot)$.

### 3.1 HARDNESS OF ESTIMATION

We first generalize the idea of constructing hard instances for learning discrete distributions on $\mathcal{A}$ (Paninski, 2008) to our nonsingleton $\mathcal{S}$ to understand the difficulty when only $(s, a)$ paris are available. We remark that the proof of Theorem 3.1 (in Appendix C.2) is the only one in this work that utilizes Assouad's method (Yu, 1997) directly.

**Theorem 3.1.** For nonempty $\mathcal{S}, \mathcal{A}$ with $|\mathcal{A}| > 1$, and $n \geq |\mathcal{S}||\mathcal{A}|/4$,

$$\inf_{\widehat{\pi} \in \widehat{\Pi}(\mathcal{D})} \sup_{\rho \times \pi^\star \in \mathcal{P}} \mathbb{E}_{(\rho \times \pi^\star)^n} \mathsf{TV}(\widehat{\pi}, \pi^\star | \rho) \gtrsim \sqrt{\frac{|\mathcal{S}||\mathcal{A}|}{n}}, \tag{3.1}$$

where $\mathcal{D} \sim (\rho \times \pi^\star)^n$, $\widehat{\Pi}(\mathcal{D})$ denotes all (possibly random) estimators mapping $\mathcal{D}$ to $\Delta(\mathcal{A}|\mathcal{S})$.

The $\sqrt{|\mathcal{S}|}$ dependence in the lower bound intuitively makes sense because the classic lower bound for $|\mathcal{S}| = 1$ is $\Omega(\sqrt{|\mathcal{A}|/n})$ and each input roughly get $n/|\mathcal{S}|$ samples when $\rho$ is distributed evenly.

### 3.2 MAXIMUM LIKELIHOOD ESTIMATION

We approximate the teacher $\pi^\star$ via minimizing the following negative log-likelihood loss:

---

[7]See Appendix B for the rigorous definitions of $\mathfrak{n}_x(X^n)$, $\mathfrak{m}_0(\nu, X^n)$, and $\mathcal{A}(\mathcal{D}, s)$.

$$\widehat{\pi}_{\mathsf{CE},\mathsf{sgl}} \in \underset{\pi\in\Delta(\mathcal{A}|\mathcal{S})}{\operatorname{argmin}} \mathsf{CE}_{\mathsf{sgl}}(\mathcal{D}) \coloneqq \underset{\pi\in\Delta(\mathcal{A}|\mathcal{S})}{\operatorname{argmin}} -\sum_{i=1}^{n}\log\pi(a_i|s_i). \tag{3.2}$$

It is possible to exactly attain the minimum 0 in (3.2). A refactoring detailed in Appendix B.1 indicates a natural relation between the hard version $\mathsf{CE}_{\mathsf{sgl}}$ and the soft version $\mathsf{CE}_{\mathsf{ful}}$, which leads to a neat closed-form solution for the optimization problem.

$$\widehat{\pi}_{\mathsf{CE},\mathsf{sgl}}(a|s) \begin{cases} = \mathfrak{n}_{(s,a)}(\mathcal{D})/\mathfrak{n}_s(\mathcal{S}(\mathcal{D})), & s\in\mathcal{S}(\mathcal{D}), \\ \in\Delta(\mathcal{A}) \text{ arbitrarily}, & \text{otherwise}. \end{cases} \tag{3.3}$$

We study the convergence behavior of $\widehat{\pi}_{\mathsf{CE},\mathsf{sgl}}$ in a fine-grained way with its proof detailed in Appendix D.2:

**Theorem 3.2.** For any $\delta\in(0,1)$, with probability at least $1-\delta$,

$$\mathsf{TV}(\widehat{\pi}_{\mathsf{CE},\mathsf{sgl}},\pi^{\star}|\rho) \lesssim \sqrt{\frac{|\mathcal{S}|(|\mathcal{A}|+\log(|\mathcal{S}|/\delta))}{n}}. \tag{3.4}$$

The upper bound in expectation $\mathbb{E}\big[\mathsf{TV}(\widehat{\pi}_{\mathsf{CE},\mathsf{sgl}},\pi^{\star}|\rho)\big] \lesssim \sqrt{|\mathcal{S}||\mathcal{A}|/n}$ is no better than (3.4) up to log factors. An instance-dependent version in expectation is

$$\mathbb{E}[\mathsf{TV}(\widehat{\pi}_{\mathsf{CE},\mathsf{sgl}},\pi^{\star}|\rho)] \lesssim \sqrt{\frac{|\mathcal{S}||\mathcal{A}|\xi(\pi^{\star})}{n}} + \frac{|\mathcal{S}|}{n}, \tag{3.5}$$

where $\xi(\pi^{\star}) \coloneqq \max_{s\in\mathcal{S}}\mathsf{dist}_{\mathsf{TV}}(\pi^{\star}(\cdot|s),\mathsf{Dirac}(\mathcal{A})) = 2\max_{s\in\mathcal{S}}\min_{a\in\mathcal{A}}(1-\pi^{\star}(a|s))$.

Thus, $\widehat{\pi}_{\mathsf{CE},\mathsf{sgl}}$ is worst-case optimal and may even have a $n^{-1}$ rate in some benign cases with $\pi^{\star}$ close enough to vertices of the simplex $\Delta(\mathcal{A})$.

## 4 TRANSFER VIA PARTIAL SLS

Besides $\mathcal{D}$, we can also access $\mathcal{R} \coloneqq \{(s_i,a_i,\pi^{\star}(a_i|s_i))\}_{i=1}^{n}$ as Partial SLs. The introduction of $\mathcal{R}$ leads to a quadratic reduction of the learning difficulty.

### 4.1 BLESSING OF GROUND TRUTH

**Theorem 4.1.** For nonempty $\mathcal{S}$, $\mathcal{A}$ with $|\mathcal{A}|>2$, and $n\geq|\mathcal{S}|(|\mathcal{A}|-1)/2-1$,

$$\inf_{\widehat{\pi}\in\widehat{\Pi}(\mathcal{D},\mathcal{R})}\sup_{\rho\times\pi^{\star}\in\mathcal{P}}\mathbb{E}_{(\rho\times\pi^{\star})^n}\mathsf{TV}(\widehat{\pi},\pi^{\star}|\rho) \gtrsim \frac{|\mathcal{S}||\mathcal{A}|}{n}, \tag{4.1}$$

where $\mathcal{D}\sim(\rho\times\pi^{\star})^n$, $\mathcal{R}=\{(s,a,\pi^{\star}(a|s)):(s,a)\in\mathcal{D}\}$, and $\widehat{\Pi}(\mathcal{D},\mathcal{R})$ denotes all (possibly random) learners mapping $(\mathcal{D},\mathcal{R})$ to $\Delta(\mathcal{A}|\mathcal{S})$.

We provide a constructive proof of Theorem 4.1 in Appendix C.3, in which we resort to the power of randomized policy so as to reveal the linear in $|\mathcal{A}|$ dependence.

### 4.2 FAILURE OF EMPIRICAL CROSS-ENTROPY LOSS

The Partial SLs $\mathcal{R}$ motivates us to define a loss $\mathsf{CE}_{\mathsf{ptl}}$ interpolating between $\mathsf{CE}_{\mathsf{sgl}}$ and $\mathsf{CE}_{\mathsf{ful}}$.

$$\widehat{\pi}_{\mathsf{CE},\mathsf{ptl}} \in \underset{\pi\in\Delta(\mathcal{A}|\mathcal{S})}{\operatorname{argmin}} \mathsf{CE}_{\mathsf{ptl}}(\mathcal{D},\mathcal{R}) \coloneqq \underset{\pi\in\Delta(\mathcal{A}|\mathcal{S})}{\operatorname{argmin}} -\sum_{i=1}^{n}\pi^{\star}(a_i|s_i)\log\pi(a_i|s_i). \tag{4.2}$$

We can obtain the following exact solution to (4.2) by another technique detailed in Appendix B.2.

$$\widehat{\pi}_{\mathsf{CE},\mathsf{ptl}}(a|s) \begin{cases} \propto \mathfrak{n}_{(s,a)}(\mathcal{D})\pi^{\star}(a|s), & s\in\mathcal{S}(\mathcal{D}), \\ \in\Delta(\mathcal{A}) \text{ arbitrarily}, & s\notin\mathcal{S}(\mathcal{D}). \end{cases} \tag{4.3}$$

The convergence analysis of $\widehat{\pi}_{\mathsf{CE,ptl}}$ crucially relies on its relationship with $\widehat{\pi}_{\mathsf{CE,sgl}}$. For any $s \in \mathcal{S}(\mathcal{D})$, $\widehat{\pi}_{\mathsf{CE,ptl}}$ can be reformulated as

$$\widehat{\pi}_{\mathsf{CE,ptl}}(a|s) = \frac{\pi^{\star}(a|s)\widehat{\pi}_{\mathsf{CE,sgl}}(a|s)}{\sum_{b \in \mathcal{A}} \pi^{\star}(b|s)\widehat{\pi}_{\mathsf{CE,sgl}}(b|s)}. \tag{4.4}$$

**Lemma 4.2.** For $|\mathcal{S}| = 1$,

$$\widehat{\pi}_{\mathsf{CE,ptl}}(\cdot|s) \xrightarrow{a.s.} [\pi^{\star}(\cdot|s)]^2 / \sum_{a \in \mathcal{A}} [\pi^{\star}(a|s)]^2,$$

under $\ell_\infty$ in $\mathbb{R}^{|\mathcal{A}|}$ if $\rho \times \pi^{\star}$ is independent of $n$.

Lemma 4.2 roughly means $\widehat{\pi}_{\mathsf{CE,ptl}} \propto (\pi^{\star})^2$ approximately as $n$ goes to infinity, which implies that small parts of $\pi^{\star}$ are underestimated and large parts of $\pi^{\star}$ are overestimated. We make this intuition technically right in Theorem 4.3, whose rigorous statement, mechanism, and proof are deferred to Appendix D.4.

**Theorem 4.3.** If $\rho \times \pi^{\star}$ does not vary with $n$, $\widehat{\pi}_{\mathsf{CE,ptl}}$ coincides with $\widehat{\pi}_{\mathsf{CE,sgl}}$ (and thus asymptotically unbiased[8]) only if $\pi^{\star}(\cdot|s) = \mathsf{Uniform}(\mathcal{A})$ or $\pi^{\star}(\cdot|s) \in \mathsf{Dirac}(\mathcal{A})$ for all $s \in \mathcal{S}$. Even for $|\mathcal{S}| = 1$, $\widehat{\pi}_{\mathsf{CE,ptl}}$ is asymptotically biased in general.

## 4.3 EMPIRICAL SQUARED ERROR LOGIT LOSS

Ba & Caruana (2014) suggests the practically promising SEL loss[9]:

$$\mathcal{L}(\pi, \pi^{\star}) = \sum_{i=1}^{n} \frac{1}{2} \sum_{a \in \mathcal{A}} [\log \pi(a|s_i) - \log \pi^{\star}(a|s_i)]^2. \tag{4.5}$$

Here we analyze the minimization of its empirical variant with *normalized* logits under the second data acquisition protocol for simplicity:

$$\mathsf{SEL}_{\mathsf{ptl}}(\mathcal{D}, \mathcal{R}) := \sum_{i=1}^{n} \frac{1}{2} [\log \pi(a_i|s_i) - \log \pi^{\star}(a_i|s_i)]^2. \tag{4.6}$$

Exact matching on the seen samples in $(\mathcal{D}, \mathcal{R})$ shows that $\widehat{\pi}_{\mathsf{SEL,ptl}} \in \arg\min_{\pi \in \Delta(\mathcal{A}|\mathcal{S})} \mathsf{SEL}_{\mathsf{ptl}}(\mathcal{D}, \mathcal{R})$

$$\Leftrightarrow \widehat{\pi}_{\mathsf{SEL,ptl}}(\cdot|s) \in \begin{cases} \{p \in \Delta(\mathcal{A}) : p(a) = \pi^{\star}(a|s), \forall a \in \mathcal{A}(\mathcal{D}, s)\}, & s \in \mathcal{S}(\mathcal{D}), \\ \Delta(\mathcal{A}) \text{ arbitrarily}, & \text{otherwise.} \end{cases} \tag{4.7}$$

The following three-fold Theorem 4.4 indicates that $\widehat{\pi}_{\mathsf{SEL,ptl}}$ converges faster than $\widehat{\pi}_{\mathsf{CE,sgl}}$ by a factor of $\sqrt{n}$ though its performance upper bound has worse dependence on $|\mathcal{S}||\mathcal{A}|$ compared with that of $\widehat{\pi}_{\mathsf{CE,sgl}}$.

**Theorem 4.4.** If $|\mathcal{S}| > 1$, for any $\delta \in (0, \min(1, (|\mathcal{S}|+2)/10))$, with probability at least $1 - \delta$,

$$\mathsf{TV}(\widehat{\pi}_{\mathsf{SEL,ptl}}, \pi^{\star}|\rho) \lesssim \frac{|\mathcal{S}|\left(|\mathcal{A}| + \sqrt{|\mathcal{A}|\log(|\mathcal{S}|/\delta)}\right)}{n} \log \frac{|\mathcal{S}|}{\delta}. \tag{4.8}$$

If $|\mathcal{S}| = 1$, for any $\delta \in (0, 1/10]$, with probability at least $1 - \delta$,

$$\mathsf{TV}(\widehat{\pi}_{\mathsf{SEL,ptl}}, \pi^{\star}|\rho) = \mathsf{TV}(\widehat{\pi}_{\mathsf{SEL,ptl}}, \pi^{\star}) \lesssim \frac{|\mathcal{A}|}{n} + \frac{\sqrt{|\mathcal{A}|}}{n} \log \frac{1}{\delta}. \tag{4.9}$$

The expected risk $\mathbb{E}\mathsf{TV}(\widehat{\pi}_{\mathsf{SEL,ptl}}, \pi^{\star}|\rho) \lesssim |\mathcal{S}||\mathcal{A}|/n$ is not polynomially tighter than (4.8) or (4.9).

---

[8]Though $\widehat{\pi}_{\mathsf{CE,ptl}}$ is not an estimator, we can discuss unbiasedness under a more general notion, i.e., for $K$ constants $\{c_i\}_{i=1}^{K}$, the random variables $\{X_{i,n}\}_{i=1}^{K}$ is called asymptotically unbiased if $X_{i,n} \to c_i$ in some mode of convergence as $n \to \infty$ for every $i \in [K]$.

[9]Practical versions of SEL often allow unnormalized logits. See Remark D.1 for more discussions.

**Remark 4.5.** Theorem 4.3 together with Theorem 4.4 manifests the advantage of employing the empirical SEL loss, which induces an alignment between the normalized logits of the learner and those of $\pi^\star$ under squared loss, over the empirical CE loss in offline distillation when the teacher is moderately reserved. A similar observation between these two style of empirical surrogate losses in online policy optimization is verfied in practice (Zhu et al., 2023b).

## 5 TRANSFER VIA SOFT LABELS

At the lightest level, the student has the extra information $\mathcal{Q} = \{(s, \pi^\star(\cdot|s) : s \in \mathcal{S}(\mathcal{D})\}$. The availability of $\mathcal{Q}$ apparently eases the transfer process, especially when the support size $|\mathcal{A}|$ of the teacher classifier is huge. Such an intuition can be precisely depicted by a $|\mathcal{A}|$-free minimax lower bound.

### 5.1 $|\mathcal{A}|$-FREE LOWER BOUND

The following lower bound roughly requires $\Omega(|\mathcal{S}|)$ burn-in cost, whose constructive proof is deferred to Appendix C.4.

**Theorem 5.1.** For $\mathcal{S}$ with $|\mathcal{S}| > 1$, $\mathcal{A}$ with $|\mathcal{A}| > 1$, and $n > |\mathcal{S}| - 1$,

$$\inf_{\widehat{\pi} \in \widehat{\Pi}(\mathcal{D}, \mathcal{Q})} \sup_{\rho \times \pi^\star \in \mathcal{P}} \mathbb{E}_{(\rho \times \pi^\star)^n} \mathsf{TV}(\widehat{\pi}, \pi^\star | \rho) \gtrsim \frac{|\mathcal{S}|}{n}, \tag{5.1}$$

where $\mathcal{D} \sim (\rho \times \pi^\star)^n$, $\mathcal{Q} = \{(s, \pi^\star(\cdot|s)) : s \in \mathcal{S}(\mathcal{D})\}$, and $\widehat{\Pi}(\mathcal{D}, \mathcal{Q})$ denotes all (possibly random) learners mapping $(\mathcal{D}, \mathcal{Q})$ to $\Delta(\mathcal{A}|\mathcal{S})$.

This setting cannot have a rate better than $n^{-1}$, which is consistent with the $n^{-1}$ rate in Theorem 4.1 since the difficulties of the later two settings (intuitively, the information provided at these two levels) should be the same when $\pi^\star(\cdot|s) \in \mathsf{Dirac}(\mathcal{A})$ for any $s \in \mathcal{S}$.

### 5.2 KULLBACK-LEIBLER DIVERGENCE MINIMIZATION

Cross-entropy loss minimization under full observation is equivalent to

$$\widehat{\pi}_{\mathsf{CE}_{\mathsf{ful}}} \in \arg\min_{\pi} \sum_{i=1}^{n} \mathsf{KL}(\pi^\star(\cdot|s_i) \| \pi(\cdot|s_i)) \Rightarrow \widehat{\pi}_{\mathsf{CE}_{\mathsf{ful}}}(\cdot|s) \begin{cases} = \pi^\star(\cdot|s), & \text{if } s \in \mathcal{S}(\mathcal{D}), \\ \in \Delta(\mathcal{A}) \text{ arbitrarily,} & \text{otherwise.} \end{cases} \tag{5.2}$$

We give the missing-mass-based proof of the matching upper bounds for $\widehat{\pi}_{\mathsf{CE}_{\mathsf{ful}}}$ in Theorem 5.2 in Appendix D.6.

**Theorem 5.2.** For any $\delta \in (0, 1/10]$, with probability at least $1 - \delta$,

$$\mathsf{TV}(\widehat{\pi}_{\mathsf{CE}_{\mathsf{ful}}}, \pi^\star | \rho) \lesssim \frac{|\mathcal{S}|}{n} + \frac{\sqrt{|\mathcal{S}|}}{n} \log \frac{1}{\delta}. \tag{5.3}$$

The upper bound $\mathbb{E}\mathsf{TV}(\widehat{\pi}_{\mathsf{CE}_{\mathsf{ful}}}, \pi^\star | \rho) \lesssim |\mathcal{S}|/n$ on the expected risk for $\widehat{\pi}_{\mathsf{CE}_{\mathsf{ful}}}$ nearly matches (5.3).

Theorem 5.2 guarantees the optimality of $\widehat{\pi}_{\mathsf{CE}_{\mathsf{ful}}}$ in that it maximally utilizes the given logits.

## 6 EXPERIMENTS

We conduct simulations to verify the intuitive performance rankings $\widehat{\pi}_{\mathsf{CE},\mathsf{sgl}} \preceq \widehat{\pi}_{\mathsf{SEL},\mathsf{ptl}} \preceq \widehat{\pi}_{\mathsf{CE}_{\mathsf{ful}}}$ given moderately large sample sizes and also numerically provide the asymptotical biasedness of $\widehat{\pi}_{\mathsf{CE},\mathsf{ptl}}$ with a finite-sample counterpart. Moreover, we design adversarial data generating distributions inspired by the information-theoretic arguments (Appendix C) for the three types of reserved teachers respectively in the non-asymptotic regime, thereby accurately exhibiting the matching convergence rates of $\widehat{\pi}_{\mathsf{CE},\mathsf{sgl}}$, $\widehat{\pi}_{\mathsf{SEL},\mathsf{ptl}}$, and $\widehat{\pi}_{\mathsf{CE}_{\mathsf{ful}}}$ in terms of $n$.

In this section, we specify a fair inductive bias due to the tabular nature: if $s \notin \mathcal{S}(\mathcal{D})$, $\widehat{\pi}(\cdot|s)$ is set to $\mathsf{Uniform}(\mathcal{A})$ for all learners; for $\widehat{\pi}_{\mathsf{SEL},\mathsf{ptl}}(\cdot|s)$, the missing mass is amortized uniformly among $\mathcal{A} \backslash \mathcal{A}(\mathcal{D}, s)$ if $s \in \mathcal{S}(\mathcal{D})$.

## 6.1 Classic Regime: Telling Learners Apart

In the classic regime, $\rho \times \pi^\star$ stays invariant no matter whether $n$ tends to infinity or not. An instance in this sense should not only expose the inferior of $\widehat{\pi}_{\mathsf{CE,ptl}}$ but also showcase the hardness of $|\mathcal{S}| > 1$, i.e., $\rho$ should be strictly bounded away from zero in $\Omega(|\mathcal{S}|)$ inputs. To these ends, we specify *Instance 0* in Appendix F. We simulate four typical realizations of it, whose estimated risks is presented in Figure 1. Each marker in Figure 1 represents the empirical mean of $\mathsf{TV}(\widehat{\pi}, \pi^\star|\rho)$ in 100 independent repeats given the corresponding sample size $n$. Any broken line in Figure 1 has nothing to do with sequential design and our experiment is purely offline. As shown in Figure 1 (a, b), either in a general case with $(|\mathcal{S}|, |\mathcal{A}|) = (100, 25)$ or for a 100-armed rewardless bandit, $\widehat{\mathbb{E}}\mathsf{TV}(\widehat{\pi}_{\mathsf{CE,ptl}}, \pi^\star|\rho)$ fails to converge but all other learners do, corroborating the asymptotically constant bias of $\widehat{\pi}_{\mathsf{CE,ptl}}$ and the consistency of the others.

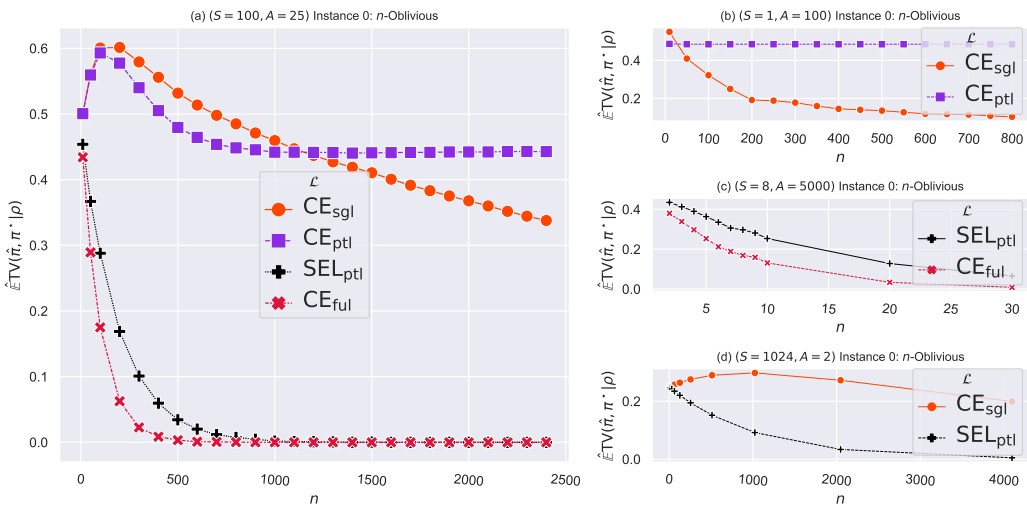

Figure 1: Estimated risks in expectation. $\rho \times \pi^\star$ does not vary with $n$.

Though Instance 0 is effectively so easy to learn for any of $\widehat{\pi}_{\mathsf{CE,sgl}}$, $\widehat{\pi}_{\mathsf{SEL,ptl}}$, and $\widehat{\pi}_{\mathsf{CE,ful}}$ that none of the three worst-case upper bounds is tightly attained, the numerical performance rankings among them in Figure 1 coincides with our intuition and theoretical analysis. The "benign"-case comparison in Figure 1 (c), where the sample sizes are small enough to make the worst-case lower bounds vacuous, still favors $\widehat{\pi}_{\mathsf{CE,ful}}$ over $\widehat{\pi}_{\mathsf{SEL,ptl}}$ in the large-$|\mathcal{A}|$ regime. Figure 1 (d) manifests that the $\sqrt{|\mathcal{S}||\mathcal{A}|}$ gap between the worst-case upper bounds for $\widehat{\pi}_{\mathsf{CE,sgl}}$ and $\widehat{\pi}_{\mathsf{SEL,ptl}}$, which is reversely dominated by the exponential rate[10] of $\widehat{\pi}_{\mathsf{SEL,ptl}}$ in this Instance 0, may not be observed in general even for large $|\mathcal{S}||\mathcal{A}|$ and small $n$.

**Remark 6.1.** Direct calculations imply that if $\rho \geq c_\mathcal{S} > 0$ for all inputs with $c_\mathcal{S}$ irrespective of $n$ when $n$ is large enough, $\mathbb{E}\mathsf{TV}(\widehat{\pi}_{\mathsf{CE,ful}}, \pi^\star|\rho)$ can decay exponentially fast, exemplified by Figure 1 (a). $\mathbb{E}\mathsf{TV}(\widehat{\pi}_{\mathsf{SEL,ptl}}, \pi^\star|\rho)$ will enjoy a similar linear convergence so long as we additionally require $\pi^\star(\cdot|s) \geq c_\mathcal{A} > 0$ for all $s \in \mathcal{S}$ and all labels with $c_\mathcal{A}$ independent of $n$ for sufficiently large $n$, again exemplified by Figure 1 (a).

## 6.2 Non-Asymptotic Regime: Illustration of Matching Rates

Instance 0 serves as an intriguing average case, but we need to design worst-case instances that may vary with $n$ (Wainwright, 2019) in the non-asymptotic regime for different data acquisition settings in order to verify the minimax optimalities. The adversarial Instance 1, 2, and 3 with their design insights are detailed in order in Appendix F for verification of matching rates at all three levels.

Since $\widehat{\pi}_{\mathsf{CE,sgl}}$, $\widehat{\pi}_{\mathsf{SEL,ptl}}$, and $\widehat{\pi}_{\mathsf{CE,ful}}$ enjoy optimal rates of order $n^{-1}$ or $n^{-0.5}$, we can manifest them using lines in a log-log plot. More generally, if some notion of $\mathtt{risk}$ has $\mathtt{risk} = \Theta(n^{\beta^\star})$ for some $\beta^\star < 0$, $\log \mathtt{risk} - \beta^\star \log n$ will be at least bounded by two straight lines on a log-log scale. We instantiate the

---

[10]See Remark 6.1 for details.

above idea for Instance 1, Instance 2, and Instance 3 in Figure 2, in which each marker represents the average of 64000 independent repeats. We also conduct linear regressions over $\mathtt{logrisk} \sim \log n$ for corresponding minimax learners and report the slope $\widehat{\beta}$ as estimated $\beta^\star$ in each subfigure of Figure 2.

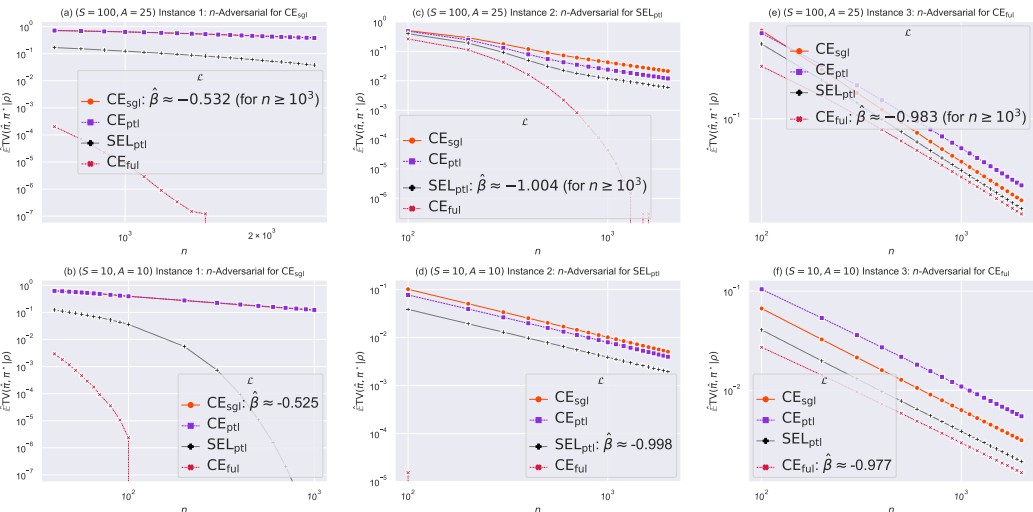

Figure 2: Estimated risks in expectation. $\rho \times \pi^\star$ scales with $n$ in a setting-specific way.

**Remark 6.2.** Regarding parameters other than $n$ as constants, the $\widehat{\beta}$'s precisely verify the worst-case rates $\mathbb{E}\mathsf{TV}(\widehat{\pi}_{\mathsf{CE,sgl}}, \pi^\star|\rho) \sim n^{-0.5}$ (Figure 2 (a, b)), $\mathbb{E}\mathsf{TV}(\widehat{\pi}_{\mathsf{SEL,ptl}}, \pi^\star|\rho) \sim n^{-1}$ (Figure 2 (c, d)), and $\mathbb{E}\mathsf{TV}(\widehat{\pi}_{\mathsf{CE_{ful}}}, \pi^\star|\rho) \sim n^{-1}$ (Figure 2 (e, f)).

**Remark 6.3.** All Instance 1, 2, and 3 happen to be too easy for $\widehat{\pi}_{\mathsf{CE,ptl}}$ to be obviously biased. This phenomenon is actually predictable because $\pi^\star(\cdot|s)$ becomes very close to $\mathsf{Uniform}(\mathcal{A})$ in Instance 1, and to some one in $\mathsf{Dirac}(\mathcal{A})$ in both Instance 2 or 3 for all inputs when $n$ is relatively large. In these cases, $\widehat{\pi}_{\mathsf{CE,ptl}}$ largely coincides with $\widehat{\pi}_{\mathsf{CE,sgl}}$ as predicted by Theorem 4.3.

**Remark 6.4.** The risks of $\widehat{\pi}_{\mathsf{SEL,ptl}}$ in Instance 1 and the risks of $\widehat{\pi}_{\mathsf{CE_{ful}}}$ in Instance 1 and 2 decay exponentially fast, which is consistent with the arguemnts in Remark 6.1, i.e., only vanishing schemes like $\pi^\star$ in Instance 2 for $\widehat{\pi}_{\mathsf{SEL,ptl}}$ and $\rho$ in Instance 3 for $\widehat{\pi}_{\mathsf{CE_{ful}}}$ can force their risks to have a polynomial decay.

**Remark 6.5.** We are able to provably explain the good performance (beyond worst cases) of $\widehat{\pi}_{\mathsf{CE,sgl}}$ in Instance 2 and 3, in which $\xi(\pi^\star) = O(n^{-1})$. (Recall the definition of $\xi(\cdot)$ in Theorem 3.2.) Thus, the instance-dependent bound in Theorem 3.2 indicates a benign-case rate of $\mathbb{E}\mathsf{TV}(\widehat{\pi}_{\mathsf{CE,sgl}}, \pi^\star|\rho) \lesssim n^{-1}$.

# 7 CONCLUSION

We embark on investigating knowledge transfer beyond pure black- or white-box regimes and settle its sample complexity, respectively; provided that the teacher can afford (1) only to act as a generative model, or (2) additionally the probabilities at sampled classes, or (3) additionally the logits conditioned on each sampled input. The theoretical analysis unveils a crucial insight that tailoring the idea of minimizing $\mathsf{CE}$ to new information acquisition scenarios may be sub-optimal in general, provably in knowledge transfer via Partial SLs. Several avenues remain to be further explored.

- Huge $\mathcal{S}$ and $\mathcal{A}$ in practice (Zeng et al., 2022; Almazrouei et al., 2023) necessitate function approximation, which is also important to our analysis itself. For example, the equivalent effectiveness of minimizing the $\mathsf{CE}$ or $\mathsf{SEL}$ loss (with Soft Labels) from an alphabet-matching perspective may be incomplete after incorporating the *approximation error*, which may consequently corroborate the empirical superior of the vanilla $\mathsf{SEL}$ loss (Ba & Caruana, 2014).

- All our upper bounds do not adapt to the student initialization strategy for unseen inputs or labels; while the student may be pre-trained in practice (Gu et al., 2023b; Jiang et al., 2023). Thus it is vital to incorporate the skillfulness of the student before transfer to the convergence analysis.

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

# A ADDITIONAL RELATED WORKS

We review key paradigms closely related to our formulation, especially KD; and point the reader to Weiss et al. (2016); Gou et al. (2021); Alyafeai et al. (2020); Zhu et al. (2023c); Wang & Yoon (2021); Tian et al. (2023) for numerous algorithms of knowledge transfer on modalities like sequences, images, graphs, etc. We restrict our survey to the single task of interest: classification. [11]

## A.1 INSPIRING PARADIGMS OF KNOWLEDGE TRANSFER IN MACHINE LEARNING

**Paradigms of Knowledge Transfer.** Knowledge transfer is not a patent of neural nets with `softmax` as their last layers. Similar ideas have realizations for ensemble of classification trees (Breiman & Shang, 1996) and even margin-based classifiers (Burges & Schölkopf, 1996). Since 2010s, there has been a line of work concentrating on the utilization of a trained teacher model and its original training set in a totally *white-box* manner with the purpose of student accuracy improvement (Hinton et al., 2015; Furlanello et al., 2018; Cho & Hariharan, 2019; Zhao et al., 2022; Romero et al., 2014; Yim et al., 2017; Huang & Wang, 2017; Park et al., 2019; Tian et al., 2019; Tung & Mori, 2019; Qiu et al., 2022). As the computation budget becomes relatively tight with respect to the scale of datasets, another line of works propose to avoid using of the full dataset for teacher pre-training during KD, and resort to architecture-specific metadata (Lopes et al., 2017), synthetic data (Nayak et al., 2019; Yin et al., 2020; Fang et al., 2022), or bootstrapping (Gu et al., 2023a) instead; which is dubbed the *data-free* approach. The sagacious vision that the teacher architecture may be agnostic or the teacher (log-)probability output may at least go through certain censorship does not receive enough attention from the community in the era of open sourcing (Orekondy et al., 2019; Wang et al., 2020; Wang, 2021; Nguyen et al., 2022). However, after the debut of closed-source and game-changing foundation models, practitioners find it plausibly nice to train their own models to purely mimic the response of these strong teachers. For example, though OpenAI only exposes transparent APIs of ChatGPT & GPT-4 (OpenAI, 2023b) to customers, there has been a line of efforts towards distilling black-box language models without even accessing the last-layer logits (Zheng et al., 2023; Wang et al., 2023b;a). Some primary results (Wang et al., 2023a) show that only letting LLaMA (Touvron et al., 2023a) mimic about 6000 carefully chosen trajectories generated by human-GPT-4 interactions can drastically boost the performance of these open-source autoregressive language models on common evaluation benchmarks (Gao et al., 2021; Li et al., 2023).

## A.2 OUR FORMULATION VERSUS PREVIOUS PARADIGMS

Motivated by the progress trend of knowledge transfer, we decouple the formulation of the input distribution $\rho$ from teacher training details and view the reference poicy $\pi^\star$ as the gold standard (ground truth), beyond just a proxy of it. In the following two paragraphs, we would like to also emphasize the distinctions between our formulations and 1) imitation learning (whose performance is measured by reward sub-optimality), or 2) the LUPI framework (Vapnik et al., 2015; Lopez-Paz et al., 2015), respectively.

*Optimality* is not explicitly defined for $\pi^\star$ in our formulation, yet we solely want to mimic (the conditional density of) the teacher, so $\pi^\star$ is dubbed the *reference policy*. This view aligns with a recent belief that foundation models are by definition "good" and in effect black-box teachers, judges, and raters of the student ones (Peng et al., 2023; Liu et al., 2023; Zheng et al., 2023). Generally speaking, optimal reward maximization is neither sufficient nor necessary for efficient knowledge transfer ($\Leftrightarrow$ accurate behavioral imitation), because an optimal policy can differ from teacher demonstrations (Ng et al., 2000), which may even be sub-optimal itself (Brown et al., 2019; Chen et al., 2021), and pure behavioral imitation can result in constant sub-optimality (Gao et al., 2023; Zhu et al., 2023a).

**Remark A.1.** This work departs from Lopez-Paz et al. (2015, Section 4.1) essentially in several ways. First, our results hold for any legal data generating distribution $\rho \times \pi^\star$ and do not need to assume data→teacher, data→student, teacher→student transfer rates manually. Second, their result crucially hinges on the assumption that the teacher learns from data faster than the student, while we have clarified our formulation of $\pi^\star$ that differs completely. Finally, their transfer speed from the teacher to the student relies on the hardness of classification of each data point, e.g., the notion of *linear separability* (Shalev-Shwartz & Ben-David, 2014), yet we consider the product space of finite domains $\mathcal{S} \times \mathcal{A}$, which does not have these issues.

---

[11]Topics related to knowledge transfer across different tasks or modalities are beyond the scope.

**Additional Notation in Appendix.** For any event $\mathcal{I}$, $\mathcal{I}^c$ denotes its complementary event. $[K] :=$ $\{1,...,K\}$ and $\overline{[K]} := \{0,...,K-1\}$ for any positive integer $K$. We set $(S,A) = (|\mathcal{S}|,|\mathcal{A}|)$ and index the input and label spaces by $(\mathcal{S},\mathcal{A}) = (\overline{[S]},[A])$ when necessary. We also use a general notion of Dirac distribution in proving the lower bounds: given any measurable space $(\mathcal{Z},\Sigma)$, we define $\mathsf{Dirac}(\mathcal{Z},z)(D) :=$ $\mathbb{1}\{z \in D\}, \forall D \in \Sigma$. We denote by $\log$ the natural logarithm and adopt the convention $0\log 0 = 0$.

# B MISSING NOTATION, DEFINITIONS, AND DERIVATIONS

**Additional Notation in Appendix.** For any event $\mathcal{I}$, $\mathcal{I}^c$ denotes its complementary event. $[K] :=$ $\{1,...,K\}$ and $\overline{[K]} := \{0,...,K-1\}$ for any positive integer $K$. We set $(S,A) = (|\mathcal{S}|,|\mathcal{A}|)$ and index the input and label spaces by $(\mathcal{S},\mathcal{A}) = (\overline{[S]},[A])$ when necessary. We also use a general notion of Dirac distribution in proving the lower bounds: given any measurable space $(\mathcal{Z},\Sigma)$, we define $\mathsf{Dirac}(\mathcal{Z},z)(D) :=$ $\mathbb{1}\{z \in D\}, \forall D \in \Sigma$. We denote by $\log$ the natural logarithm and adopt the convention $0\log 0 = 0$.

**Definition B.1.** For $n$ *i.i.d.* samples $X^n$ drawn from a distribution $\nu$ over an alphabet $\mathcal{X}$, the number of occurrences of $x$ is denoted by $\mathfrak{n}_x(X^n) := \sum_{i=1}^n \mathbb{1}\{X_i = x\}$, upon which we further measure the portion of $\mathcal{X}$ never observed in $X^n$ by the missing mass

$$\mathfrak{m}_0(\nu,X^n) := \sum_{x \in \mathcal{S}} \nu(x)\mathbb{1}\{\mathfrak{n}_x(X^n) = 0\}. \tag{B.1}$$

It is worth mentioning that, $\mathcal{D}$, $\mathcal{S}(\mathcal{D})$, $\mathcal{A}(\mathcal{D})$, and $\mathcal{A}(\mathcal{D},s), \forall s \in \mathcal{S}$ are treated as *multisets* when fed into functionals like $\mathfrak{m}_0(\nu,\cdot)$, $\mathfrak{n}_x(\cdot)$, or the cardinality operator $|\cdot|$, for example, $|\mathcal{A}(\mathcal{D},s)| = \mathfrak{n}_s(\mathcal{S}(\mathcal{D}))$; while in set operations like $\mathcal{S} \backslash \mathcal{S}(\mathcal{D})$ or under the summation sign like $\sum_{s \in \mathcal{S}(\mathcal{D})}$, where they act as ranges of enumeration, we slightly abuse the notations for simplicity to refer to their *deduplicated* counterparts.

**Definition B.2.** All the $\mathfrak{n}_s(\mathcal{S}(\mathcal{D}))$ labels in $\mathcal{D}$ mapped from $s_i = s$ form a multiset

$$\mathcal{A}(\mathcal{D},s) := \{a \in \mathcal{A} : \text{for all } (x,a) \in \mathcal{D} \text{ s.t. } x = s\}.$$

## B.1 MAXIMUM LIKELIHOOD ESTIMATION

$$\widehat{\pi}_{\mathsf{CE},\mathsf{sgl}} \in \operatorname*{argmin}_{\pi \in \Delta(\mathcal{A}|\mathcal{S})} \mathsf{CE}_{\mathsf{sgl}}(\mathcal{D})$$

$$= \operatorname*{argmin}_{\pi \in \Delta(\mathcal{A}|\mathcal{S})} \sum_{i=1}^n \mathsf{CE}_{\mathsf{sgl}}(s_i,a_i;\pi) = \operatorname*{argmin}_{\pi \in \Delta(\mathcal{A}|\mathcal{S})} -\sum_{s \in \mathcal{S}} \sum_{a \in \mathcal{A}} \mathfrak{n}_{(s,a)}(\mathcal{D})\log(a|s)$$

$$= \operatorname*{argmin}_{\pi \in \Delta(\mathcal{A}|\mathcal{S})} -\sum_{s \in \mathcal{S}(\mathcal{D})} \mathfrak{n}_s(\mathcal{S}(\mathcal{D})) \underbrace{\sum_{a \in \mathcal{A}} \frac{\mathfrak{n}_{(s,a)}(\mathcal{D})}{\mathfrak{n}_s(\mathcal{S}(\mathcal{D}))}\log\pi(a|s)}_{-\mathsf{CE}_{\mathsf{ful}}\left(\mathfrak{n}_{(s,\cdot)}(\mathcal{D})/\mathfrak{n}_s(\mathcal{S}(\mathcal{D}))\|\pi(\cdot|s)\right)}. \tag{B.2}$$

Noticing that (B.2) is the summation of the cross-entropy between $\mathfrak{n}_{(s,\cdot)}(\mathcal{D})/\mathfrak{n}_s(\mathcal{S}(\mathcal{D}))$ and $\pi(\cdot|s)$ weighted by $\mathfrak{n}_s(\mathcal{S}(\mathcal{D}))$ over $\mathcal{S}(\mathcal{D})$, we figure out the explicit solution of $\widehat{\pi}_{\mathsf{CE},\mathsf{sgl}}$ as

$$\widehat{\pi}_{\mathsf{CE},\mathsf{sgl}}(a|s) \begin{cases} = \mathfrak{n}_{(s,a)}(\mathcal{D})/\mathfrak{n}_s(\mathcal{S}(\mathcal{D})), & s \in \mathcal{S}(\mathcal{D}), \\ \in \Delta(\mathcal{A}) \text{ arbitrarily}, & \text{otherwise}. \end{cases}$$

## B.2 EMPIRICAL CROSS-ENTROPY LOSS

$$\widehat{\pi}_{\mathsf{CE},\mathsf{ptl}} \in \operatorname*{argmin}_{\pi \in \Delta(\mathcal{A}|\mathcal{S})} \mathsf{CE}_{\mathsf{ptl}}(\mathcal{D},\mathcal{R}) = \operatorname*{argmin}_{\pi \in \Delta(\mathcal{A}|\mathcal{S})} -\sum_{s \in \mathcal{S}(\mathcal{D})} Z_s \sum_{a \in \mathcal{A}} \frac{\mathfrak{n}_{(s,a)}(\mathcal{D})\pi^\star(a|s)}{Z_s}\log\pi(a|s),$$

where $Z_s := \sum_{a \in \mathcal{A}} \mathfrak{n}_{(s,a)}(\mathcal{D})\pi^\star(a|s)$. Therefore, by the same cross-entropy minimization argument, the explicit solution is

$$\widehat{\pi}_{\mathsf{CE},\mathsf{ptl}}(a|s) \begin{cases} = \begin{cases} \mathfrak{n}_{(s,a)}(\mathcal{D})\pi^\star(a|s)/Z_s, & (s,a) \in \mathcal{D}, \\ 0, & s \in \mathcal{S}(\mathcal{D}), a \notin \mathcal{A}(\mathcal{D},s), \end{cases} & \propto \mathfrak{n}_{(s,a)}(\mathcal{D})\pi^\star(a|s); \\ \in \Delta(\mathcal{A}) \text{ arbitrarily}, & s \notin \mathcal{S}(\mathcal{D}). \end{cases}$$

## C INFORMATION-THEORETIC ARGUMENTS

### C.1 ADDITIONAL NOTATION IN APPENDIX C

We write $\mathsf{KL}(\pi\|\pi'|\lambda) := \mathbb{E}_{x\sim\lambda}[\mathsf{KL}(\pi(\cdot|x)\|\pi'(\cdot|x))]$ for $\lambda\in\Delta(\mathcal{S})$ and $\pi,\pi'\in\Delta(\mathcal{A}|\mathcal{S})$ likewise for alphabets $\mathcal{S},\mathcal{A}$ to have notations concise. The values of $\triangle$ in the proofs of Theorem 3.1, Theorem 4.1, and Theorem 5.1 are different under the same notation. A similar logic applies to the values of $\pi_\tau$ in the proofs of Theorem 3.1 and Theorem 4.1.

### C.2 PROOF OF THEOREM 3.1

*Proof.* We fix $\rho$ to be $\mathsf{Uniform}(\mathcal{S})$ and define a loss function over $\Delta(\mathcal{A}|\mathcal{S})\times\Delta(\mathcal{A}|\mathcal{S})$ as

$$l(\pi,\pi') := \mathsf{TV}(\pi,\pi'|\rho) = \frac{1}{S}\sum_{s=0}^{S-1}\mathsf{TV}(\pi(\cdot|s),\pi'(\cdot|s)). \tag{C.1}$$

Obviously, $(\Delta(\mathcal{A}|\mathcal{S}),l)$ becomes a metric space. Without loss of generality, suppose $A$ is even, we decompose $l$ as

$$d_{sA/2+j}(\pi,\pi') := l_{s,j}(\pi,\pi') := \frac{|\pi(2j-1|s)-\pi'(2j-1|s)|+|\pi(2j|s)-\pi'(2j|s)|}{2S}, \tag{C.2}$$

$$l(\pi,\pi') = \sum_{s=0}^{S-1}\sum_{j=1}^{A/2}l_{s,j}(\pi,\pi') = \sum_{i=1}^{SA/2}d_i(\pi,\pi'). \tag{C.3}$$

Inspired by Paninski's construction (Paninski, 2008), we define $\pi_\tau$ as

$$\pi_\tau(2j-1|s) = \frac{1+\tau_{sA/2+j}\triangle}{A}, \pi_\tau(2j|s) = \frac{1-\tau_{sA/2+j}\triangle}{A}, \forall(s,j)\in\overline{[S]}\times\left[\frac{A}{2}\right]; \tag{C.4}$$

where $\tau\in\{-1,+1\}^{AS/2}$ and $\triangle$ is to be specified later. For any $\tau\sim_i\tau'$, i.e., any pair in $\{-1,+1\}^{AS/2}$ that differs only in the $i$-th coordinate, the construction of $\pi_\tau$ leads to

$$d_i(\pi_\tau,\pi_{\tau'}) = \frac{2\triangle}{SA}. \tag{C.5}$$

We thereby refer $\tau\sim\tau'$ to any pair in $\{-1,+1\}^{AS/2}$ that differs only in one coordinate and obtain

$$\begin{aligned}
\mathsf{LHS\ of\ (3.1)} &\geq \inf_{\widehat{\pi}\in\widehat{\Pi}(\mathcal{D})}\sup_{\substack{\tau\in\{-1,+1\}^{AS/2}\\\rho=\mathsf{Uniform}(\mathcal{S})}}\mathbb{E}_{(\rho\times\pi_\tau)^n}l(\widehat{\pi},\pi_\tau)\\
&\geq \frac{AS}{2}\cdot\frac{2\triangle/(SA)}{2}\min_{\tau\sim\tau'}(1-\mathsf{TV}((\rho\times\pi_\tau)^n,(\rho\times\pi_{\tau'})^n))\\
&\geq \frac{\triangle}{4}\min_{\tau\sim\tau'}\exp(-\mathsf{KL}((\rho\times\pi_\tau)^n\|(\rho\times\pi_{\tau'})^n))\\
&= \frac{\triangle}{4}\min_{\tau\sim\tau'}\exp(-n\mathsf{KL}(\rho\times\pi_\tau\|\rho\times\pi_{\tau'})) = \frac{\triangle}{4}\min_{\tau\sim\tau'}\exp(-n\mathsf{KL}(\pi_\tau\|\pi_{\tau'}|\rho))\\
&= \frac{\triangle}{4}\exp\left(-\frac{n}{SA}\cdot2\triangle\log\frac{1+\triangle}{1-\triangle}\right)\\
&\geq \frac{\triangle}{4}\exp\left(-8\frac{n}{SA}\triangle^2\right), \tag{C.6}
\end{aligned}$$

where

- the second inequality is by Assouad's lemma (Yu, 1997, Lemma 2),

- the third inequality holds due to a variant (Lemma E.2) of the Bretagnolle–Huber inequality,

- the first equality holds due to the decomposable property of KL (Tsybakov, 2009, Section 2.4),

- the second equality follows from a basic property of $f$-divergence (Polyanskiy & Wu, 2022, Proposition 7.2.4),

- the last equality is by the definition of $\pi_\tau$ in (C.4) and $\rho = \mathsf{Uniform}(\mathcal{S})$,

- the last inequality derives from $\log(1+x) \le x, x > 0$ and an additional constraint $\triangle \le 0.5$ we impose.

The assignment $\triangle = 0.25\sqrt{SA/n}$ with $n \ge SA/4$ is a feasible choice for inequality (C.6) to hold, whose RHS further equals to

$$\frac{\exp(-0.5)}{16}\sqrt{\frac{SA}{n}}.$$

$\square$

## C.3   Proof of Theorem 4.1

*Proof.* Without loss of generality, we assume $A > 2$ is odd, then for a fixed $\triangle := 1$, which does NOT vary with $S$, $A$, or $n$ in THIS proof, we define $\Pi := \left\{\pi_\tau : \tau \in \{-1, +1\}^{AS/2}\right\}$, where

$$\left.\begin{aligned}
\pi_\tau(2j-1|s) &:= \frac{S(1+\tau_{sA/2+j}\triangle)}{2(n+1)}, \\
\pi_\tau(2j|s) &:= \frac{S(1-\tau_{sA/2+j}\triangle)}{2(n+1)}; \\
\pi_\tau(A|s) &:= 1 - \frac{S}{2}\cdot\frac{A-1}{n+1}.
\end{aligned}\right\} (s,j) \in \overline{[S]} \times \left[\frac{A-1}{2}\right]. \tag{C.7}$$

To get a lower bound on a Bayes risk, we design a prior over $\mathcal{P}$ as

$$\Lambda := \mathsf{Dirac}(\Delta(\mathcal{S}), \mathsf{Uniform}(\mathcal{S})) \times \Gamma, \tag{C.8}$$

where

$$\Gamma := \mathsf{Uniform}(\Pi).$$

Intuitively speaking, for any $\rho \times \pi$ sampled from $\Lambda$, $\rho$ must be uniform over $\mathcal{S}$ and $\pi$ must be some $\pi_\tau$ with $\tau$ uniformly distributed over $\{-1, +1\}^{SA/2}$.[12] Therefore, if we let $\Lambda(\mathcal{D}, \mathcal{R})$ be the corresponding posterior over $\mathcal{P}$ conditioned on $(\mathcal{D}, \mathcal{R})$ and $\Gamma(\mathcal{D}, \mathcal{R})$ be the marginal posterior over $\Delta(\mathcal{A}|\mathcal{S})$, we can by the definition of $\Gamma$ and $\pi_\tau$ obtain $\Lambda(\mathcal{D}, \mathcal{R}) = \mathsf{Dirac}(\Delta(\mathcal{S}), \mathsf{Uniform}(\mathcal{S})) \times \Gamma(\mathcal{D}, \mathcal{R})$ where for any $(s,a) \in \{0,...,S-1\} \times \{1,...,A-1\}$ and $\pi \sim \Gamma(\mathcal{D}, \mathcal{R})$, by the Bayes rule,

$$\begin{cases} \Gamma(\mathcal{D},\mathcal{R})[\pi(a|s) = \pi^\star(a|s)] = 1, (s,a) \in \mathcal{D} \text{ or } (s, \mathsf{Buddy}(a)) \in \mathcal{D}; \\ \Gamma(\mathcal{D},\mathcal{R})\left[\pi(a|s) = \frac{S(1+\triangle)}{2(n+1)}\right] = \Gamma(\mathcal{D},\mathcal{R})\left[\pi(a|s) = \frac{S(1-\triangle)}{2(n+1)}\right] = \frac{1}{2}, \text{otherwise}; \end{cases} \tag{C.9}$$

where (recall that $A > 2$ is assumed to be odd) we define the "Buddy" for $a \in [A-1] = \{1,...,A-1\}$ as

$$\mathsf{Buddy}(a) := \begin{cases} a-1, & a \text{ is even}; \\ a+1, & a \text{ is odd}. \end{cases} \tag{C.10}$$

The intuition behind (C.9) is that if $\pi \sim \Gamma(\mathcal{D}, \mathcal{R})$, the marginal posterior of $\pi(a|s)$ for any seen $(s,a)$ in $\mathcal{D}$ must be a Dirac concentrated at $\pi^\star(a|s)$ and by the design of $\Pi = \{\pi_\tau\}$, $\pi(a|s)$ is also determined if $(s, \mathsf{Buddy}(a)) \in \mathcal{D}$.[13] Note that the last label, $A$, is designed to be ignored, which we do not consider in both (C.9) and the following argument driven by Fubini's theorem. We define a event $\mathcal{E}_\mathcal{D}(s,a)$ for every $(s,a)$ with $a < A$ as

$$\mathcal{E}_\mathcal{D}(s,a) = (s,a) \in \mathcal{D} \text{ or } (s, \mathsf{Buddy}(a)) \in \mathcal{D}.$$

Next, we can apply Fubini's theorem to the Bayes risk with $\Lambda$ as its prior.

$$\mathbb{E}_{\rho \times \pi^\star \sim \Lambda}\left[\mathbb{E}_{(\rho \times \pi^\star)^n} \mathsf{TV}(\widehat{\pi}, \pi^\star|\rho)\right]$$

---

[12]To be technically rigorous, we say the prior for $\rho$ over $\Delta(\mathcal{S})$ and the prior for $\pi^\star$ over $\Delta(\mathcal{A}|\mathcal{S})$ are assigned *independently*, similar assignment also appears in the proof of Theorem 5.1.

[13]Rigorously speaking, $\pi^\star(a|s)$ in (C.9) refers to some concrete realization to some $\pi_\tau(a|s)$, which is collected by $\mathcal{R}$. It is a slight abuse of notation and a similar one appears in (C.15) in the proof of Theorem 5.1, too.

$$= \frac{1}{S}\sum_{s\in\mathcal{S}}\mathbb{E}_{\pi^\star\sim\Gamma}\mathbb{E}_{\mathcal{D}\sim(\rho\times\pi^\star)^n}\mathbb{E}_{\pi\sim\Gamma(\mathcal{D},\mathcal{Q})}\mathsf{TV}(\widehat{\pi}(\cdot|s),\pi(\cdot|s))$$

$$\geq \frac{1}{2S}\sum_{s\in\mathcal{S}}\mathbb{E}_{\pi^\star\sim\Gamma}\mathbb{E}_{\mathcal{D}\sim(\rho\times\pi^\star)^n}\sum_{a<A}\mathbb{E}_{\pi\sim\Gamma(\mathcal{D},\mathcal{Q})}|\widehat{\pi}(a|s)-\pi(a|s)|$$

$$= \frac{1}{2S}\sum_{s\in\mathcal{S}}\mathbb{E}_{\pi^\star\sim\Gamma}\mathbb{E}_{\mathcal{D}\sim(\rho\times\pi^\star)^n}\sum_{a<A}\{$$
$$\mathbb{E}_{\pi\sim\Gamma(\mathcal{D},\mathcal{Q})}[|\widehat{\pi}(a|s)-\pi(a|s)||\mathcal{E}_{\mathcal{D}}(s,a)]\mathbb{E}[\mathbb{1}\{\mathcal{E}_{\mathcal{D}}(s,a)\}|\mathcal{D}]$$
$$+\mathbb{E}_{\pi\sim\Gamma(\mathcal{D},\mathcal{Q})}[|\widehat{\pi}(a|s)-\pi(a|s)||\mathcal{E}_{\mathcal{D}}^c(s,a)]\mathbb{E}[\mathbb{1}\{\mathcal{E}_{\mathcal{D}}^c(s,a)\}|\mathcal{D}]$$
$$\}$$

$$\geq \frac{1}{2S}\sum_{s\in\mathcal{S}}\mathbb{E}_{\pi^\star\sim\Gamma}\mathbb{E}_{\mathcal{D}\sim(\rho\times\pi^\star)^n}\sum_{a<A}\mathbb{E}_{\pi\sim\Gamma(\mathcal{D},\mathcal{Q})}[|\widehat{\pi}(a|s)-\pi(a|s)||\mathcal{E}_{\mathcal{D}}^c(s,a)]\mathbb{E}[\mathbb{1}\{\mathcal{E}_{\mathcal{D}}^c(s,a)\}|\mathcal{D}]$$

$$= \frac{1}{2S}\sum_{s\in\mathcal{S}}\mathbb{E}_{\pi^\star\sim\Gamma}\mathbb{E}_{\mathcal{D}\sim(\rho\times\pi^\star)^n}\sum_{a<A}\mathbb{E}[\mathbb{1}\{\mathcal{E}_{\mathcal{D}}^c(s,a)\}|\mathcal{D}]\cdot\{$$
$$\frac{1}{2}\left|\widehat{\pi}(a|s)-\frac{S(1+\triangle)}{2(n+1)}\right|+\frac{1}{2}\left|\widehat{\pi}(a|s)-\frac{S(1-\triangle)}{2(n+1)}\right|$$
$$\}$$

$$\geq \frac{1}{2S}\sum_{s\in\mathcal{S}}\mathbb{E}_{\pi^\star\sim\Gamma}\mathbb{E}_{\mathcal{D}\sim(\rho\times\pi^\star)^n}\sum_{a<A}\mathbb{E}[\mathbb{1}\{\mathcal{E}_{\mathcal{D}}^c(s,a)\}|\mathcal{D}]\frac{1}{2}\cdot\frac{S}{n+1} \tag{C.11}$$

$$= \frac{1}{4(n+1)}\sum_{s,a:a<A}\mathbb{P}\{(s,a)\notin\mathcal{D} \text{ and } (s,\mathsf{Buddy}(a))\notin\mathcal{D}\} \tag{C.12}$$

$$= \frac{1}{4(n+1)}\sum_{s,a:a<A}\left(1-\rho(s)\cdot\frac{S}{n+1}\right)^n = \frac{1}{4(n+1)}\sum_{s,a:a<A}\left(1-\frac{1}{S}\cdot\frac{S}{n+1}\right)^n$$

$$= \frac{1}{4(n+1)}\sum_{s,a:a<A}\left(1-\frac{1}{n+1}\right)^n$$

$$\geq \frac{S(A-1)}{4e(n+1)}\gtrsim\frac{SA}{n}, \tag{C.13}$$

where the inequality in (C.11) holds because of the triangle inequality and $\triangle=1$ by design, and the equality in (C.12) holds because for any possible value of $\pi^\star$ a priori, i.e., any $\pi_\tau$, $\mathbb{P}\{\mathcal{E}_{\mathcal{D}}^c(s,a)\}$ is always $(1-1/(n+1))^n$ by the design of $\Pi=\{\pi_\tau\}$; the penultimate and the last equality hold due to the fact that the distribution of $\rho$ is always $\mathsf{Dirac}(\Delta(\mathcal{S}),\mathsf{Uniform}(\mathcal{S}))$ no matter whether a priori or a posteriori (conditioned on $(\mathcal{D},\mathcal{R})$). Since the minimax risk is bounded from below by the worst-case Bayes risk (Polyanskiy & Wu, 2022, Theorem 28.1), the proof is completed. $\square$

## C.4 Proof of Theorem 5.1

*Proof.* We assign $\xi=1/(n+1)$ and design a $p\in\Delta(\mathcal{S})$ as $p(0)=1-S-1/(n+1)$ and $p=1/(n+1)$ for all other inputs following Rajaraman et al. (2020, Figure 1 (b)). Then it suffices to get a lower bound for a Bayes risk given a prior over $\mathcal{P}$, which we design as

$$\Lambda_3 := \mathsf{Dirac}(\Delta(\mathcal{S}),p)\times\Gamma_3, \tag{C.14}$$

where

$$\Gamma_3 := \mathsf{Uniform}(\Pi_{\mathrm{det}}),\Pi_{\mathrm{det}} := \{\pi\in\Delta(\mathcal{A}|\mathcal{S}):\pi(\cdot|s)\in\mathsf{Dirac}(\mathcal{A}),\forall s\in\mathcal{S}\}.$$

Intuitively speaking, $\pi^\star$ is uniformly distributed over all deterministic policies, which indicates the marginal prior distribution of $\pi^\star(\cdot|s)$ for any $s\in\mathcal{S}$ is

$$\Gamma_3[\pi(\cdot|s)=\mathsf{Dirac}(\mathcal{A},a)]=\frac{1}{A},\forall a\in\mathcal{A}.$$

We abbreviate the corresponding posterior of $\Lambda_3$ (resp. $\Gamma_3$) conditioned on $(\mathcal{D},\mathcal{Q})$ as $\Lambda_3(\mathcal{D},\mathcal{Q})$ (resp. $\Gamma_3(\mathcal{D},\mathcal{Q})$), which by definition implies $\Lambda_3(\mathcal{D},\mathcal{Q})=\mathsf{Dirac}(\Delta(\mathcal{S}),p)\times\Gamma_3(\mathcal{D},\mathcal{Q})$ and by the Bayes

formula implies that for any $s \in \mathcal{S}$ and $\pi \sim \Gamma_3(\mathcal{D}, \mathcal{Q})$,

$$
\begin{cases}
\Gamma_3(\mathcal{D}, \mathcal{Q})[\pi(\cdot|s) = \pi^\star(\cdot|s)] & = 1, s \in \mathcal{S}(\mathcal{D}); \\
\Gamma_3(\mathcal{D}, \mathcal{Q})[\pi(\cdot|s) = \mathsf{Dirac}(\mathcal{A}, a)] & = 1/A, s \in \mathcal{S} \backslash \mathcal{S}(\mathcal{D}).
\end{cases}
\tag{C.15}
$$

Without loss of generality, we assume $A > 1$ is even and then by Fubini's theorem,

$$
\mathbb{E}_{\rho \times \pi^\star \sim \Lambda_3} \big[ \mathbb{E}_{(\rho \times \pi^\star)^n} \mathsf{TV}(\widehat{\pi}, \pi^\star | \rho) \big]
$$

$$
= \sum_{s \in \mathcal{S}} \rho(s) \mathbb{E}_{\pi^\star \sim \Gamma_3} \mathbb{E}_{\mathcal{D} \sim (\rho \times \pi^\star)^n} \mathbb{E}_{\pi \sim \Gamma_3(\mathcal{D}, \mathcal{Q})} \mathsf{TV}(\widehat{\pi}(\cdot|s), \pi(\cdot|s))
$$

$$
= \sum_{s \in \mathcal{S}} \rho(s) \mathbb{E}_{\pi^\star \sim \Gamma_3} \mathbb{E}_{\mathcal{D} \sim (\rho \times \pi^\star)^n} \big\{ \mathbb{E}_{\pi \sim \Gamma_3(\mathcal{D}, \mathcal{Q})} \big[ \mathsf{TV}(\widehat{\pi}(\cdot|s), \pi(\cdot|s)) \big| s \in \mathcal{S}(\mathcal{D}) \big] \mathbb{E}[\mathbb{1}(s \in \mathcal{S}(\mathcal{D})) | \mathcal{D}]
$$

$$
+ \mathbb{E}_{\pi \sim \Gamma_3(\mathcal{D}, \mathcal{Q})} \big[ \mathsf{TV}(\widehat{\pi}(\cdot|s), \pi(\cdot|s)) \big| s \notin \mathcal{S}(\mathcal{D}) \big] \mathbb{E}[\mathbb{1}(s \notin \mathcal{S}(\mathcal{D})) | \mathcal{D}] \big\}
$$

$$
\geq \sum_{s \in \mathcal{S}} \rho(s) \mathbb{E}_{\pi^\star \sim \Gamma_3} \mathbb{E}_{\mathcal{D} \sim (\rho \times \pi^\star)^n} \big\{ \mathbb{E}[\mathbb{1}(s \notin \mathcal{S}(\mathcal{D})) | \mathcal{D}] \mathbb{E}_{\pi \sim \Gamma_3(\mathcal{D}, \mathcal{Q})} \big[ \mathsf{TV}(\widehat{\pi}(\cdot|s), \pi(\cdot|s)) \big| s \notin \mathcal{S}(\mathcal{D}) \big] \big\}
$$

$$
= \sum_{s \in \mathcal{S}} \rho(s) \mathbb{E}_{\pi^\star \sim \Gamma_3} \mathbb{E}_{(\rho \times \pi^\star)^n} \Big\{ \mathbb{E}[\mathbb{1}(s \notin \mathcal{S}(\mathcal{D})) | \mathcal{D}] \cdot \frac{1}{A} \sum_{a \in \mathcal{A}} \mathsf{TV}(\widehat{\pi}(\cdot|s), \mathsf{Dirac}(\mathcal{A}, a)) \Big\},
$$

$$
= \sum_{s \in \mathcal{S}} \rho(s) \mathbb{E}_{\pi^\star \sim \Gamma_3} \mathbb{E}_{(\rho \times \pi^\star)^n} \big\{
$$

$$
\mathbb{E}[\mathbb{1}(s \notin \mathcal{S}(\mathcal{D})) | \mathcal{D}] \cdot \frac{1}{A} \sum_{a=0}^{A/2-1} [\mathsf{TV}(\widehat{\pi}(\cdot|s), \mathsf{Dirac}(\mathcal{A}, a)) + \mathsf{TV}(\widehat{\pi}(\cdot|s), \mathsf{Dirac}(\mathcal{A}, a+A/2))]
$$

$$
\big\}
$$

$$
\geq \sum_{s \in \mathcal{S}} \rho(s) \mathbb{P}(s \notin \mathcal{S}(\mathcal{D})) \frac{1}{A} \sum_{a=0}^{A/2-1} \mathsf{TV}(\mathsf{Dirac}(\mathcal{A}, a), \mathsf{Dirac}(\mathcal{A}, a+A/2))
$$

$$
= \sum_{s \in \mathcal{S}} \rho(s) \mathbb{P}(s \notin \mathcal{S}(\mathcal{D})) \frac{1}{A} \cdot \frac{A}{2} = \frac{1}{2} \sum_{s \in \mathcal{S}} \rho(s) \mathbb{P}(s \notin \mathcal{S}(\mathcal{D})),
\tag{C.16}
$$

where the last inequality holds due to the triangle inequality of TV. Therefore,

$$
\text{LHS of (C.16)} \geq 0.5 \sum_{s \in \mathcal{S}} \rho(s) \mathbb{P}(s \notin \mathcal{S}(\mathcal{D})) = 0.5 \sum_{s \in \mathcal{S}} \rho(s)(1 - \rho(s))^n
$$

$$
\geq \frac{S-1}{2(n+1)} \left(1 - \frac{1}{n+1}\right)^n \geq \frac{S-1}{2e(n+1)} \gtrsim \frac{S}{n},
\tag{C.17}
$$

where the second inequality is by only considering the $S - 1$ inputs with mass $1/(n+1)$. Since the minimax risk is bounded from below by the worst-case Bayes risk, the proof is completed. $\square$

# D    ARGUMENTS FOR SPECIFIC LEARNERS

## D.1    ADDITIONAL DEFINITIONS IN APPENDIX D

In this section we denote the MLE of $\rho$ by

$$
\widehat{\rho}(\cdot) := \frac{\mathfrak{n}_{(\cdot)}(\mathcal{S}(\mathcal{D}))}{n}.
\tag{D.1}
$$

The event $B_{s,i}$ defined as follows will be used in the proofs of Theorem 3.2 and Theorem 4.4.

$$
B_{s,i} := \{\mathfrak{n}_s(\mathcal{S}(\mathcal{D})) = i\}, \forall (s,i) \in \mathcal{S} \times \overline{[n+1]}.
\tag{D.2}
$$

### D.2 PROOF OF THEOREM 3.2

#### D.2.1 PROOF OF THE HIGH-PROBABILITY BOUND

*Proof.* For $|\mathcal{S}| > 1$, we define

$$u_s := \mathfrak{n}_s(\mathcal{S}(\mathcal{D}))\mathsf{TV}(\widehat{\pi}_{\mathsf{CE,sgl}}(\cdot|s), \pi^\star(\cdot|s)), \forall s \in \mathcal{S}.$$

We decompose the LHS of (3.4) as

$$
\begin{aligned}
\mathsf{LHS} &= \sum_{s \in \mathcal{S}} \left( \frac{\mathfrak{n}_s(\mathcal{S}(\mathcal{D}))}{n} + \rho(s) - \widehat{\rho}(s) \right) \mathsf{TV}(\widehat{\pi}_{\mathsf{CE,sgl}}(\cdot|s), \pi^\star(\cdot|s)) \\
&\leq \sum_{s \in \mathcal{S}} \frac{u_s}{n} + \sum_{s \in \mathcal{S}} |\rho(s) - \widehat{\rho}(s)| \mathsf{TV}(\widehat{\pi}_{\mathsf{CE,sgl}}(\cdot|s), \pi^\star(\cdot|s)) \\
&\leq \underbrace{\frac{1}{n} \sum_{s \in \mathcal{S}} u_s + 2\mathsf{TV}(\rho, \widehat{\rho})}_{(i)},
\end{aligned}
\tag{D.3}
$$

where the first inequality is by triangle inequality and the second one holds due to the boundedness of TV. We define another two types of events to bound $(i)$ in (D.3):

$$D_s := \left\{ u_s \leq \sqrt{\frac{\mathfrak{n}_s(\mathcal{S}(\mathcal{D}))}{2} \left( |\mathcal{A}|\log 2 + \log \frac{|\mathcal{S}|+1}{\delta} \right)} \right\}, \forall s \in \mathcal{S}; \tag{D.4}$$

$$E := \left\{ 2\mathsf{TV}(\rho, \widehat{\rho}) \leq \sqrt{\frac{2}{n} \left( |\mathcal{S}|\log 2 + \log \frac{|\mathcal{S}|+1}{\delta} \right)} \right\}. \tag{D.5}$$

Notice that $\mathbb{P}(D_s^c|B_{s,0}) = 0$ by the definition of $B_{s,i}$ in (D.2) and for any $i > 0$, $\mathbb{P}(D_s^c|B_{s,i}) \leq \delta/(|\mathcal{S}|+1), \forall s \in \mathcal{S}$ by Lemma E.4; thus by the law of total probability,

$$\mathbb{P}(D_s) = \sum_{i=0}^{n} \mathbb{P}(D_s|B_{s,i})\mathbb{P}(B_{s,i}) \geq \left( 1 - \frac{\delta}{|\mathcal{S}|+1} \right) \sum_{i=0}^{n} \mathbb{P}(B_{s,i}) = 1 - \frac{\delta}{|\mathcal{S}|+1}, \forall s \in \mathcal{S}. \tag{D.6}$$

Also noticing that $\mathbb{P}(E^c) \leq \delta/(|\mathcal{S}|+1)$ by Lemma E.4, we apply a union bound over $E^c$ and $\{D_s^c\}_{s \in \mathcal{S}}$ for $(i)$ in (D.3) to conclude that with probability at least $1 - \delta$,

$$(i) \leq \sqrt{\frac{|\mathcal{A}|\log 2 + \log((|\mathcal{S}|+1)/\delta)}{2}} \cdot \underbrace{\frac{\sum_{s \in \mathcal{S}} \sqrt{\mathfrak{n}_s(\mathcal{S}(\mathcal{D}))}}{n}}_{\heartsuit} + \sqrt{\frac{2}{n} \left( |\mathcal{S}|\log 2 + \log \frac{|\mathcal{S}|+1}{\delta} \right)}. \tag{D.7}$$

By the Cauchy-Schwarz inequality,

$$\heartsuit \text{ in (D.7)} \leq \frac{1}{n} \sqrt{|\mathcal{S}| \sum_{s \in \mathcal{S}} \mathfrak{n}_s(\mathcal{S}(\mathcal{D}))} = \sqrt{\frac{|\mathcal{S}|}{n}}. \tag{D.8}$$

Substituting (D.8) back to the RHS of (D.7) yields the conclusion. The case of $|\mathcal{S}| = 1$ follows from Lemma E.4. □

#### D.2.2 PROOF OF THE WORST-CASE UPPER BOUND IN EXPECTATION

*Proof.* Taking expectation on both sides of (D.3) yields

$$
\begin{aligned}
\mathbb{E}[\mathsf{TV}(\widehat{\pi}_{\mathsf{CE,sgl}}, \pi^\star|\rho)] &\leq \frac{1}{n} \sum_{s \in \mathcal{S}} \mathbb{E}u_s + \sqrt{\frac{|\mathcal{S}|}{n}} \\
&= \frac{1}{n} \sum_{s \in \mathcal{S}} \mathbb{E}\left[ \underbrace{\mathfrak{n}_s(\mathcal{S}(\mathcal{D}))\mathbb{E}[\mathsf{TV}(\widehat{\pi}_{\mathsf{CE,sgl}}(\cdot|s), \pi^\star(\cdot|s))|\mathfrak{n}_s(\mathcal{S}(\mathcal{D}))]}_{=:\widetilde{u}_s} \right] + \sqrt{\frac{|\mathcal{S}|}{n}},
\end{aligned}
\tag{D.9}
$$

where the inequality holds due to Lemma E.3. For every $s$, we trivially have

$$\widetilde{u}_s \leq \frac{\sqrt{|\mathcal{A}| \mathfrak{n}_s(\mathcal{S}(\mathcal{D}))}}{2} \tag{D.10}$$

if conditioned on $B_{s,0}$. If otherwise conditioned on $B_{s,0}^c$, we still have

$$\widetilde{u}_s \leq \frac{\sqrt{|\mathcal{A}| \mathfrak{n}_s(\mathcal{S}(\mathcal{D}))}}{2}, \tag{D.11}$$

where the inequality follows from Lemma E.3. Therefore, substituting (D.10) and (D.11) back to (D.9) gives

$$\text{LHS of (D.9)} \leq \frac{\sqrt{|\mathcal{A}|}}{2n} \sum_{s \in \mathcal{S}} \mathbb{E}\sqrt{\mathfrak{n}_s(\mathcal{S}(\mathcal{D}))} + \sqrt{\frac{|\mathcal{S}|}{n}} \leq \sqrt{\frac{|\mathcal{A}|}{2n}} \sum_{s \in \mathcal{S}} \sqrt{\rho(s)} + \sqrt{\frac{|\mathcal{S}|}{n}}$$

$$\lesssim \sqrt{\frac{|\mathcal{S}||\mathcal{A}|}{n}},$$

where the first inequality follows from the law of total expectation with respect to $B_{s,0}$ and $B_{s,0}^c$, the second inequality follows from Jensen's inequality together with the definition of $\mathfrak{n}_s(\mathcal{S}(\mathcal{D}))$, and the last inequality is by the Cauchy-Schwarz inequality. $\square$

### D.2.3 PROOF OF THE INSTANCE-DEPEDENT UPPER BOUND IN EXPECTATION

*Proof.* We define the set of the numbers of occurences of all inputs as

$$N_{\mathcal{S}} := \{\mathfrak{n}_s(\mathcal{S}(\mathcal{D})) : s \in \mathcal{S}\}. \tag{D.12}$$

Then we decompose the LHS of (3.5) as

$$\sum_{s \in \mathcal{S}} \rho(s) \mathbb{E}[\mathsf{TV}(\widehat{\pi}_{\mathsf{CE},\mathsf{sgl}}(\cdot|s), \pi^\star(\cdot|s))]$$

$$= \mathbb{E}\Big[$$

$$\sum_{s \in \mathcal{S}(\mathcal{D})} \rho(s) \mathbb{E}\Big[\mathsf{TV}(\widehat{\pi}_{\mathsf{CE},\mathsf{sgl}}(\cdot|s), \pi^\star(\cdot|s)) \Big| N_{\mathcal{S}}\Big]$$

$$+ \sum_{s \in \mathcal{S} \setminus \mathcal{S}(\mathcal{D})} \rho(s) \mathbb{E}\Big[\mathsf{TV}(\widehat{\pi}_{\mathsf{CE},\mathsf{sgl}}(\cdot|s), \pi^\star(\cdot|s)) \Big| N_{\mathcal{S}}\Big]$$

$$\Big]$$

$$\leq \underbrace{\mathbb{E}\Big[\sum_{s \in \mathcal{S}(\mathcal{D})} \rho(s) \mathbb{E}\Big[\mathsf{TV}(\widehat{\pi}_{\mathsf{CE},\mathsf{sgl}}(\cdot|s), \pi^\star(\cdot|s)) \Big| \mathfrak{n}_s(\mathcal{S}(\mathcal{D}))\Big]\Big]}_{I_1} + \underbrace{\mathbb{E}\mathfrak{m}_0(\rho, \mathcal{S}(\mathcal{D}))}_{I_2}, \tag{D.13}$$

where the inequality is by the definition and boundedness of $\mathsf{TV}(\widehat{\pi}_{\mathsf{CE},\mathsf{sgl}}(\cdot|s), \pi^\star(\cdot|s))$. We divide $I_1$ and $I_2$ so at to conquer them as follows.

**Bounding $I_1$.** For every $s \in \mathcal{S}$, we define

$$I_1(s) := \mathbb{E}\Big[\mathsf{TV}(\widehat{\pi}_{\mathsf{CE},\mathsf{sgl}}(\cdot|s), \pi^\star(\cdot|s)) \Big| \mathfrak{n}_s(\mathcal{S}(\mathcal{D}))\Big].$$

Then we can bound $I_1(s)$ by Jensen's inequality for $s \in \mathcal{S}(\mathcal{D})$:

$$I_1(s) = \frac{1}{2} \sum_{a \in \mathcal{A}} \mathbb{E}\Big[|\widehat{\pi}_{\mathsf{CE},\mathsf{sgl}}(a|s) - \pi^\star(a|s)| \Big| \mathfrak{n}_s(\mathcal{S}(\mathcal{D}))\Big]$$

$$\leq \frac{1}{2} \sum_{a \in \mathcal{A}} \sqrt{\mathbb{E}\Big[\frac{(\mathfrak{n}_s(\mathcal{S}(\mathcal{D}))\widehat{\pi}_{\mathsf{CE},\mathsf{sgl}}(a|s) - \mathfrak{n}_s(\mathcal{S}(\mathcal{D}))\pi^\star(a|s))^2}{[\mathfrak{n}_s(\mathcal{S}(\mathcal{D}))]^2} \Big| \mathfrak{n}_s(\mathcal{S}(\mathcal{D}))\Big]}$$

$$= \frac{1}{2} \sum_{a \in \mathcal{A}} \sqrt{\frac{\pi^\star(a|s)(1 - \pi^\star(a|s))}{\mathfrak{n}_s(\mathcal{S}(\mathcal{D}))}}, \tag{D.14}$$

where the last equality holds due to the observation that

$$\mathfrak{n}_s(\mathcal{S}(\mathcal{D}))\widehat{\pi}_{\mathsf{CE},\mathsf{sgl}}(a|s)\big|\mathfrak{n}_s(\mathcal{S}(\mathcal{D}))\sim\mathsf{Binomial}(\mathfrak{n}_s(\mathcal{S}(\mathcal{D})),\pi^\star(a|s)).$$

Therefore, we can bound the summation inside the expectation of $I_1$ as

$$
\begin{aligned}
\sum_{s\in\mathcal{S}(\mathcal{D})}\rho(s)I_1(s)&=\frac{1}{2}\sum_{a\in\mathcal{A}}\sum_{s\in\mathcal{S}(\mathcal{D})}\sqrt{\rho(s)}\sqrt{\rho(s)\frac{\pi^\star(a|s)(1-\pi^\star(a|s))}{\mathfrak{n}_s(\mathcal{S}(\mathcal{D}))}}\\
&\leq\frac{1}{\sqrt{2}}\sum_{a\in\mathcal{A}}\sum_{s\in\mathcal{S}}\sqrt{\rho(s)}\sqrt{\rho(s)\frac{\pi^\star(a|s)(1-\pi^\star(a|s))}{1+\mathfrak{n}_s(\mathcal{S}(\mathcal{D}))}}\\
&\leq\frac{1}{\sqrt{2}}\sum_{a\in\mathcal{A}}\sqrt{\sum_{s\in\mathcal{S}}\rho(s)\frac{\pi^\star(a|s)(1-\pi^\star(a|s))}{1+\mathfrak{n}_s(\mathcal{S}(\mathcal{D}))}},
\end{aligned}
\tag{D.15}
$$

where the first inequality holds due to $\mathfrak{n}_s(\mathcal{S}(\mathcal{D}))\geq 1,\forall s\in\mathcal{S}(\mathcal{D})$ and the last inequality is by the Cauchy-Schwarz inequality. Substituting (D.15) into $I_1=\mathbb{E}[\sum_{s\in\mathcal{S}(\mathcal{D})}\rho(s)I_1(s)]$ gives

$$
\begin{aligned}
I_1&\leq\frac{1}{\sqrt{2}}\sum_{a\in\mathcal{A}}\sqrt{\sum_{s\in\mathcal{S}}\pi^\star(a|s)(1-\pi^\star(a|s))\mathbb{E}\frac{\rho(s)}{1+\mathfrak{n}_s(\mathcal{S}(\mathcal{D}))}}\\
&\leq\frac{1}{\sqrt{2}}\sum_{a\in\mathcal{A}}\sqrt{\sum_{s\in\mathcal{S}}\frac{\pi^\star(a|s)(1-\pi^\star(a|s))}{n+1}}\\
&\leq\frac{1}{\sqrt{2(n+1)}}\sum_{a\in\mathcal{A}}\sqrt{\sum_{s\in\mathcal{S}}\min(\pi^\star(a|s),1-\pi^\star(a|s))}\\
&\leq\frac{1}{\sqrt{2(n+1)}}\sqrt{|\mathcal{A}|\sum_{a\in\mathcal{A}}\sum_{s\in\mathcal{S}}\min(\pi^\star(a|s),1-\pi^\star(a|s))}\\
&\leq\sqrt{\frac{|\mathcal{S}||\mathcal{A}|}{2(n+1)}}\cdot\sqrt{\underbrace{\max_{s\in\mathcal{S}}\sum_{a\in\mathcal{A}}\min(\pi^\star(a|s),1-\pi^\star(a|s))}_{\widetilde{\xi}(\pi^\star)}},
\end{aligned}
\tag{D.16}
$$

where the first inequality is by Jensen's inequality, the second inequality derives from Lemma E.8, and the penultimate inequality holds due to the Cauchy-Schwarz inequality. $\widetilde{\xi}(\pi^\star)$ in (D.16) can be further bounded from above by

$$
\begin{aligned}
\max_{s\in\mathcal{S}}\min_{b\in\mathcal{A}}\left(1-\pi^\star(b|s)+\sum_{a:a\neq b}\pi^\star(a|s)\right)&=\max_{s\in\mathcal{S}}\min_{b\in\mathcal{A}}\mathsf{TV}(\pi^\star(\cdot|s),\mathsf{Dirac}(\mathcal{A},b))\\
&=\max_{s\in\mathcal{S}}\mathsf{dist}_{\mathsf{TV}}(\pi^\star(\cdot|s),\mathsf{Dirac}(\mathcal{A}))=\xi(\pi^\star).
\end{aligned}
$$

To sum up, $I_1\lesssim\sqrt{\xi(\pi^\star)|\mathcal{S}||\mathcal{A}|n^{-1}}$.

**Bounding $I_2$.** Explicit calculation yields

$$I_2=\sum_{s\in\mathcal{S}}\rho(s)(1-\rho(s))^n\leq\frac{4|\mathcal{S}|}{9n}\lesssim\frac{|\mathcal{S}|}{n},$$

where the inequality follows from Lemma E.5. $\qquad\square$

### D.3 PROOF OF LEMMA 4.2

*Proof.* Since $|\mathcal{S}|=1$, we omit the conditioning on $s\in\mathcal{S}$ for brevity in this proof. By Lemma E.1,

$$\widehat{\pi}_{\mathsf{CE},\mathsf{sgl}}\xrightarrow{a.s.}\pi^\star.$$

Therefore, applying the continuous mapping theorem ([Durrett](#), [2019](#), Theorem 3.2.10) to (4.4) gives

$$\widehat{\pi}_{\mathsf{CE},\mathsf{ptl}}(a) \xrightarrow{a.s.} \frac{[\pi^{\star}(a)]^2}{\sum_{b \in \mathcal{A}}[\pi^{\star}(b)]^2}$$

*uniformly* for every $a \in \mathcal{A}$.

$\square$

## D.4 PROOF OF THEOREM 4.3

*Proof.* By (3.3) and (4.3), $\widehat{\pi}_{\mathsf{CE},\mathsf{sgl}}(\cdot|s) \propto \mathfrak{n}_s(\mathcal{S}(\mathcal{D}))$ and $\widehat{\pi}_{\mathsf{CE},\mathsf{ptl}}(\cdot|s) \propto \mathfrak{n}_s(\mathcal{S}(\mathcal{D}))\pi^{\star}(\cdot|s)$ for any $s \in \mathcal{S}(\mathcal{D})$. Therefore, the solution set of $\widehat{\pi}_{\mathsf{CE},\mathsf{ptl}}$ coincides with that of $\widehat{\pi}_{\mathsf{CE},\mathsf{sgl}}$ only if $\pi^{\star} = \mathsf{Uniform}(\mathcal{A})$ or $\pi^{\star} \in \mathsf{Dirac}(\mathcal{A})$; otherwise Lemma 4.2 implies that

$$\liminf_{n \to \infty} \mathsf{TV}(\widehat{\pi}_{\mathsf{CE},\mathsf{ptl}}, \pi^{\star}|\rho) > 0 \text{ almost surely,}$$

and we thus rigorously justify *the* $\Omega(1)$ in Table 1 with probability one. $\square$

## D.5 PROOF OF THEOREM 4.4

### D.5.1 PROOF OF THE HIGH-PROBABILITY BOUNDS

*Proof.* For $|\mathcal{S}| > 1$, we define

$$v_s := \mathfrak{n}_s(\mathcal{S}(\mathcal{D}))\mathsf{TV}(\widehat{\pi}_{\mathsf{SEL},\mathsf{ptl}}(\cdot|s), \pi^{\star}(\cdot|s))$$

and decompose LHS of (4.8) into three terms as

$$\mathsf{LHS} \leq \sum_{s \in \mathcal{S}} \widehat{\rho}(s)\mathsf{TV}(\widehat{\pi}_{\mathsf{SEL},\mathsf{ptl}}(\cdot|s), \pi^{\star}(\cdot|s)) + \sum_{s \in \mathcal{S}} |\rho(s) - \widehat{\rho}(s)|\mathsf{TV}(\widehat{\pi}_{\mathsf{SEL},\mathsf{ptl}}(\cdot|s), \pi^{\star}(\cdot|s)) \quad \text{(D.17)}$$

$$\leq \frac{1}{n}\sum_{s \in \mathcal{S}} v_s + \underbrace{\frac{1}{n}\sum_{s \in \mathcal{S}(\mathcal{D})}\left|\frac{\rho(s)}{\widehat{\rho}(s)} - 1\right|v_s}_{(ii)} + \overbrace{\sum_{s \in \mathcal{S} \backslash \mathcal{S}(\mathcal{D})}\rho(s)}^{\mathfrak{m}_0(\rho, \mathcal{S}(\mathcal{D})), \text{ matches Definition B.1}}, \quad \text{(D.18)}$$

where $\widehat{\rho}$ is defined in (D.1) and the second inequality follows from $\widehat{\rho}(s) = 0, \forall s \in \mathcal{S} \backslash \mathcal{S}(\mathcal{D})$ along with the boundedness of TV. We additionally define two kinds of events to bound $(ii)$ in (D.18).

$$\check{D}_s = \left\{v_s \leq \frac{4}{9}|\mathcal{A}| + 3\sqrt{|\mathcal{A}|\log\frac{|\mathcal{S}|+2}{\delta}}\right\}, \forall s \in \mathcal{S};$$

$$\check{E} = \left\{\mathfrak{m}_0(\rho, \mathcal{S}(\mathcal{D})) \leq \frac{4|\mathcal{S}|}{9n} + \frac{3\sqrt{|\mathcal{S}|}}{n}\log\frac{|\mathcal{S}|+2}{\delta}\right\}.$$

Recall Definition B.2 for $\mathcal{A}(\mathcal{D},s)$, if $\mathfrak{n}_s(\mathcal{S}(\mathcal{D})) > 0$, by (4.7),

$$2\mathsf{TV}(\widehat{\pi}_{\mathsf{SEL},\mathsf{ptl}}(\cdot|s), \pi^{\star}(\cdot|s)) = \sum_{a \in \mathcal{A}}|\widehat{\pi}_{\mathsf{SEL},\mathsf{ptl}}(a|s) - \pi^{\star}(a|s)|$$

$$= \sum_{a \in \mathcal{A} \backslash \mathcal{A}(\mathcal{D},s)}|\widehat{\pi}_{\mathsf{SEL},\mathsf{ptl}}(a|s) - \pi^{\star}(a|s)| \leq 2\mathfrak{m}_0(\pi^{\star}(\cdot|s), \mathcal{A}(\mathcal{D},s)), \quad \text{(D.19)}$$

where the inequality holds due to triangle inequality. Consequently,

$$v_s \leq \mathfrak{n}_s(\mathcal{S}(\mathcal{D}))\mathfrak{m}_0(\pi^{\star}(\cdot|s), \mathcal{A}(\mathcal{D},s)), \forall s \in \mathcal{S}(\mathcal{D});$$

to which we apply Lemma E.7 to obtain

$$\mathbb{P}(\check{D}_s^c|B_{s,i}) \leq \frac{\delta}{|\mathcal{S}|+2}, \forall i > 0;$$

where $B_{s,i}$ is defined in (D.2). Also noticing that $\mathbb{P}(\check{D}_s^c|B_{s,0})=0$ by definition, we can control $\mathbb{P}(\check{D}_s)$ by

$$\mathbb{P}(\check{D}_s)=\sum_i^n \mathbb{P}(\check{D}_s|B_{s,i})\mathbb{P}(B_{s,i}) \geq \left(1-\frac{\delta}{|\mathcal{S}|+2}\right)\sum_i^n \mathbb{P}(B_{s,i})=1-\frac{\delta}{|\mathcal{S}|+2},\forall s\in\mathcal{S}.$$

Since $\mathbb{P}(\check{E}) \geq 1-\delta/(|\mathcal{S}|+2)$ by Lemma E.7, we apply a union bound over $\check{E}^c$ and $\{\check{D}_s^c\}_{s\in\mathcal{S}}$ for $(ii)$ in (D.18) to conclude that with probability at least $1-(|\mathcal{S}|+1)\delta/(|\mathcal{S}|+2)$,

$$(ii) \leq \frac{4|\mathcal{A}|+3\sqrt{|\mathcal{A}|\log((|\mathcal{S}|+2)/\delta)}}{9n} \cdot \left(|\mathcal{S}|+\overbrace{\sum_{s\in\mathcal{S}(\mathcal{D})}\underbrace{\left\lfloor\left|\frac{\rho(s)}{\widehat{\rho}(s)}-1\right|\right\rfloor}_{=:o_s}}^{\blacktriangle}\right) \tag{D.20}$$

$$+\frac{4|\mathcal{S}|}{9n}+\frac{3\sqrt{|\mathcal{S}|}}{n}\log\frac{|\mathcal{S}|+2}{\delta}.$$

We further decompose $\blacktriangle$ in (D.20) in a *pragmatically* tight enough way as

$$\blacktriangle \leq |\mathcal{S}|+\sum_{s\in\mathcal{S}(\mathcal{D})}\frac{\rho(s)}{\widehat{\rho}(s)}=|\mathcal{S}|+\sum_{s\in\mathcal{S}(\mathcal{D})}\frac{n\rho(s)}{\mathfrak{n}_s(\mathcal{S}(\mathcal{D}))}$$

$$=|\mathcal{S}|+\sum_{s\in\mathcal{S}(\mathcal{D})}\frac{2n\rho(s)}{\mathfrak{n}_s(\mathcal{S}(\mathcal{D}))+\mathfrak{n}_s(\mathcal{S}(\mathcal{D}))} \leq |\mathcal{S}|+2\sum_{s\in\mathcal{S}(\mathcal{D})}\frac{n\rho(s)}{\mathfrak{n}_s(\mathcal{S}(\mathcal{D}))+1}$$

$$\leq |\mathcal{S}|+2\sum_{s\in\mathcal{S}}\frac{n\rho(s)}{\mathfrak{n}_s(\mathcal{S}(\mathcal{D}))+1}$$

$$=|\mathcal{S}|+\underbrace{\sum_{s\in\bar{\mathcal{S}}}\frac{n\rho(s)}{\mathfrak{n}_s(\mathcal{S}(\mathcal{D}))+1}}_{\bar{\blacktriangle}}+\overbrace{\sum_{s\in\widetilde{\mathcal{S}}}\underbrace{\frac{n\rho(s)}{\mathfrak{n}_s(\mathcal{S}(\mathcal{D}))+1}}_{r_s}}^{\widetilde{\blacktriangle}}, \tag{D.21}$$

where $\bar{\mathcal{S}}$ and $\widetilde{\mathcal{S}}$ are defined as

$$\bar{\mathcal{S}}:=\left\{s\in\mathcal{S}:0<\rho(s)<\frac{\log(|\mathcal{S}|(|\mathcal{S}|+2)/\delta)}{n}\cdot\frac{200}{99}\right\}, \tag{D.22}$$

$$\widetilde{\mathcal{S}}:=\left\{s\in\mathcal{S}:\rho(s)\geq\frac{\log(|\mathcal{S}|(|\mathcal{S}|+2)/\delta)}{n}\cdot\frac{200}{99}\right\}. \tag{D.23}$$

By the definition of $\bar{\mathcal{S}}$, all $s$'s in $\bar{\mathcal{S}}$ have small enough $\rho(s)$, and thus $\bar{\blacktriangle}$ in (D.21) can be trivially bounded from above, i.e.,

$$\bar{\blacktriangle} \leq \frac{200}{99}|\mathcal{S}|\log\frac{|\mathcal{S}|(|\mathcal{S}|+2)}{\delta}. \tag{D.24}$$

For each $s\in\widetilde{\mathcal{S}}$, we define

$$\eta_s:=\sqrt{\frac{2\log(|\mathcal{S}|(|\mathcal{S}|+2)/\delta)}{n\rho(s)}},$$

then by the definition of $\widetilde{\mathcal{S}}$, $1-\eta_s\geq 0.1$. Therefore, noticing that $\mathfrak{n}_s(\mathcal{S}(\mathcal{D}))\sim\text{Binomial}(n,\rho(s))$, we can apply Corollary E.10 to each $r_s$ in (D.21) to conclude that for every $s\in\widetilde{\mathcal{S}}$, with probability at least $1-\delta/(|\mathcal{S}|(|\mathcal{S}|+2))$,

$$\frac{r_s}{n\rho(s)} \leq \frac{1}{(1-\eta_s)n\rho(s)+1} \leq \frac{1}{0.1n\rho(s)+1} \leq \frac{10}{n\rho(s)}, \tag{D.25}$$

which followed by a union bound argument within $\widetilde{\mathcal{S}}$ yields that with probability at least $1-\delta/(|\mathcal{S}|+2)$,

$$\widetilde{\blacktriangle} \text{ in (D.21)} \le 10|\mathcal{S}|. \tag{D.26}$$

We then denote by $\widetilde{E}$ the event conditioned on which (D.26) holds and denote by $\dot{E}$ the event conditioned on which (D.20) holds. A union bound argument over $\widetilde{E}^c$ and $\dot{E}^c$ shows that with probability at least $1-\delta$, the LHS of (4.8) is bounded from above by

$$\frac{4|\mathcal{A}|+3\sqrt{|\mathcal{A}|\log(((|\mathcal{S}|+2)/\delta)}}{9n} \cdot \left(12|\mathcal{S}|+\frac{200}{99}|\mathcal{S}|\log\frac{|\mathcal{S}|(|\mathcal{S}|+2)}{\delta}\right)+\frac{4|\mathcal{S}|}{9n}+\frac{3\sqrt{|\mathcal{S}|}}{n}\log\frac{|\mathcal{S}|+2}{\delta}.$$

For $|\mathcal{S}|=1$, invoking Lemma E.7 for (D.19) to draw the conclusion.

$\square$

### D.5.2 Proof of the Upper Bound in Expectation

*Proof.* Substituting (D.19) into (D.17) yields an upper bound for $\mathbb{E}\mathsf{TV}(\widehat{\pi}_{\mathsf{SEL,ptl}},\pi^\star|\rho)$ as

$$\frac{1}{n}\sum_{s\in\mathcal{S}}\mathbb{E}v_s+\sum_{s\in\mathcal{S}}\mathbb{E}\left[\underbrace{|\rho(s)-\widehat{\rho}(s)|\mathbb{E}[\mathfrak{m}_0(\pi^\star(\cdot|s),\mathcal{A}(\mathcal{D},s))|\mathfrak{n}_s(\mathcal{S}(\mathcal{D}))]}_{\widetilde{v}_s}\right]. \tag{D.27}$$

Each $\mathbb{E}v_s$ in (D.27) can be bounded from above via (D.19) by

$$\mathbb{E}\left[\underbrace{\mathfrak{n}_s(\mathcal{S}(\mathcal{D}))\mathbb{E}[\mathfrak{m}_0(\pi^\star(\cdot|s),\mathcal{A}(\mathcal{D},s))|\mathfrak{n}_s(\mathcal{S}(\mathcal{D}))]}_{\bar{v}_s}\right]. \tag{D.28}$$

Conditioned on $B_{s,0}^c$, invoking Lemma E.5 to conclude

$$\bar{v}_s \le \frac{4|\mathcal{A}|}{9}. \tag{D.29}$$

The above inequality trivially holds if otherwise conditioned on $B_{s,0}$. Similarly, we always have

$$\widetilde{v}_s \le \rho(s)\frac{|\mathcal{A}|}{\mathfrak{n}_s(\mathcal{S}(\mathcal{D}))+1}+\frac{4|\mathcal{A}|}{9n}. \tag{D.30}$$

Substituting (D.29) back to (D.28) gives

$$\frac{1}{n}\sum_{s\in\mathcal{S}}\mathbb{E}v_s \lesssim \frac{|\mathcal{S}||\mathcal{A}|}{n}. \tag{D.31}$$

By Lemma E.8, substituing (D.30) and (D.31) back to (D.27) yields

$$\mathbb{E}\mathsf{TV}(\widehat{\pi}_{\mathsf{SEL,ptl}},\pi^\star|\rho) \lesssim \frac{|\mathcal{S}||\mathcal{A}|}{n}+|\mathcal{A}|\sum_{s\in\mathcal{S}}\mathbb{E}\frac{\rho(s)}{\mathfrak{n}_s(\mathcal{S}(\mathcal{D}))+1} \lesssim \frac{|\mathcal{S}||\mathcal{A}|}{n}. \tag{D.32}$$

$\square$

**Remark D.1.** The empirical variant of vanilla SEL in our second case actually match the log probability, which is by definition normalized. Analyzing an unnormalized version, which is more relevant to the matching the logits in practice (Ba & Caruana, 2014; Kim et al., 2021), in the second setting may call for new techniques. Also, some preliminary results on the empirical side manifest the difference between minimizing forward KL and reverse KL in scenarios related to our last setting (Jiang et al., 2019; Gu et al., 2023b; Agarwal et al., 2023), whose analysis are left are future work.

### D.6 PROOF OF THEOREM 5.2

*Proof.* Since TV is bounded from above by 1 and $\mathsf{TV}(\widehat{\pi}_{\mathsf{CE_{ful}}}(\cdot|s), \pi^\star(\cdot|s)) = 0, \forall s \in \mathcal{S}(\mathcal{D})$,

$$\mathsf{LHS} = \sum_{s \in \mathcal{S} \setminus \mathcal{S}(\mathcal{D})} \rho(s) \mathsf{TV}(\widehat{\pi}_{\mathsf{CE_{ful}}}(\cdot|s), \pi^\star(\cdot|s)) \leq \sum_{s \in \mathcal{S}} \rho(s) \mathbb{1}\{s \notin \mathcal{S}(\mathcal{D})\} =: M. \tag{D.33}$$

Noticing that $M$ realizes Definition B.1 to $\mathfrak{m}_0(\rho, \mathcal{S}(\mathcal{D}))$, we invoke Lemma E.7 to get

$$M \leq \mathbb{E}M + \frac{3\sqrt{|\mathcal{S}|}\log(1/\delta)}{n} \leq \frac{4|\mathcal{S}|}{9n} + \frac{3\sqrt{|\mathcal{S}|}\log(1/\delta)}{n}. \tag{D.34}$$

Substituting (D.34) back to (D.33) finishes the proof. $\qquad\square$

## E AUXILIARY LEMMAS

In contrast with other non-asymptotic tools below, we must assume the mass $p$ does not vary with $n$ in the asymptotic guarantee Lemma E.1.

**Lemma E.1.** Let $p$ be a probability mass function over an alphabet $\mathcal{S}$, whose empirical estimation from $X_1, ..., X_n \overset{i.i.d.}{\sim} p$ is $p_n(\cdot) := \sum_{i=1}^{n} \mathbb{1}\{X_i = \cdot\}/n$, then

$$p_n \xrightarrow{a.s.} p,$$

where the almost surely convergence is defined under the $\ell_\infty$ metric in $\mathbb{R}^{|\mathcal{S}|}$.

*Proof.* Without loss of generality, we assume $\mathcal{S} = [|\mathcal{S}|]$; thereby inducing $p(x) = F(x) - F(x-1)$ for $x \in [|\mathcal{S}|]$, where $F(x) = \mathbb{P}(X \leq x)$ is the distribution function of $X \sim p$. Similarly, $p_n(x) = F_n(x) - F_n(x-1)$ for

$$F_n(\cdot) = \sum_{i=1}^{n} \frac{\mathbb{P}(X_i \leq \cdot)}{n}.$$

Therefore,

$$\max_{x \in [|\mathcal{S}|]} |p_n(x) - p(x)| \leq \sup_{x \in \mathbb{R}} |F_n(x) - F(x) - (F_n(x-1) - F(x-1))|$$

$$\leq 2\sup_{x \in \mathbb{R}} |F_n(x) - F(x)| =: 2\|F_n - F\|_\infty.$$

The proof is thus completed by invoking the Glivenko-Cantelli Theorem (Van der Vaart, 2000, Theorem 19.1). $\qquad\square$

### E.1 BOUNDING TV FROM ABOVE

**Lemma E.2** (Bretagnolle–Huber inequality (Bretagnolle & Huber, 1979))**.** If $P$ and $Q$ are two probability measures on the same measurable space, then

$$\mathsf{TV}(P, Q) \leq 1 - \frac{1}{2}\exp(-\mathsf{KL}(P\|Q)).$$

**Lemma E.3.** If $a_1, ..., a_n \overset{i.i.d.}{\sim} \pi \in \Delta(\mathcal{A})$, whose MLE is $\widehat{\pi} = \widehat{\pi}(a_1, ..., a_n)$; and $|\mathcal{A}| < \infty$, then

$$\mathbb{E}\mathsf{TV}(\widehat{\pi}, \pi) \leq \frac{1}{2}\sqrt{\frac{|\mathcal{A}|}{n}}.$$

*Proof.* We reproduce the proof of this standard result here for completeness.

$$\mathsf{LHS} = \frac{1}{2}\sum_{a \in \mathcal{A}} \mathbb{E}|\widehat{\pi}(a) - \pi(a)| \leq \frac{1}{2}\sum_{a \in \mathcal{A}} \sqrt{\mathbb{E}(\widehat{\pi}(a) - \pi(a))^2}$$

$$= \frac{1}{2}\sum_{a \in \mathcal{A}} \sqrt{\frac{1}{n^2}\mathrm{Var}(n\widehat{\pi}(a))} = \frac{1}{2\sqrt{n}}\sum_{a \in \mathcal{A}} \sqrt{\pi(a)(1 - \pi(a))}$$

$$\leq \frac{1}{2\sqrt{n}}\sum_{a\in\mathcal{A}}\sqrt{\pi(a)}\leq\frac{1}{2}\sqrt{\frac{|\mathcal{A}|}{n}},$$

where the first inequality is by Jensen's inequality, the third equality holds due to $n\widehat{\pi}(a)\sim\mathsf{Binomail}(n,\pi(a))$, and the last inequality is by the Cauchy-Schwarz inequaity. $\qquad\square$

**Lemma E.4.** Under the same setting as Lemma E.3, for any $\delta\in(0,1)$, with probability at least $1-\delta$,

$$\mathsf{TV}(\widehat{\pi},\pi)\leq\sqrt{\frac{|\mathcal{A}|\log 2+\log(1/\delta)}{2n}}.$$

*Proof.* This is a straightforward corollary of the Bretagnolle-Huber-Carol inequality (van der Vaart & Wellner, 1996, Proposition A.6.6) based on the relationship between TV and $\ell_1$. $\qquad\square$

### E.2 Missing Mass Analysis

Observations like Lemma E.5 are key and common in the analysis of learning from finite and static datasets (Rajaraman et al., 2020; Rashidinejad et al., 2021).

**Lemma E.5.** For all $x\in[0,1],n>0$, $x(1-x)^n\leq(4/9)n$.

*Proof.* By taking the derivative w.r.t. $x$,

$$\max_{x\in[0,1]}\mathsf{LHS}=\left(\frac{n}{n+1}\right)^{n+1}\frac{1}{n}\leq\frac{1}{n}\lim_{n\to\infty}(1-\frac{1}{n+1})^{n+1}=\frac{1}{en}\leq\frac{4}{9n}.$$

$\qquad\square$

**Lemma E.6** (Rajaraman et al. 2020, Theorem A.2). Given a distribution $\nu$ on an alphabet $\mathcal{S}$ and $n$ *i.i.d.* samples $X^n\overset{i.i.d.}{\sim}\nu$, then for any $\delta\in(0,1/10]$, with probability at least $1-\delta$,

$$\mathfrak{m}_0(\nu,X^n)-\mathbb{E}[\mathfrak{m}_0(\nu,X^n)]\leq\frac{3\sqrt{|\mathcal{S}|}\log(1/\delta)}{n}.$$

**Lemma E.7.** Under the same setting as Lemma E.6, for any $\delta\in(0,1/10]$, with probability at least $1-\delta$,

$$\mathfrak{m}_0(\nu,X^n)\leq\frac{4|\mathcal{S}|}{9n}+\frac{3\sqrt{|\mathcal{S}|}\log(1/\delta)}{n}.$$

*Proof.*

$$\mathbb{E}\mathfrak{m}_0(\nu,X^n)=\sum_{x\in\mathcal{S}}\nu(x)\mathbb{E}\mathbb{1}\{x\notin X^n\}=\sum_{x\in\mathcal{S}}\nu(x)\mathbb{P}(x\notin X^n)$$
$$=\sum_{x\in\mathcal{S}}\nu(x)(1-\nu(x))^n\leq\frac{4|\mathcal{S}|}{9n}, \tag{E.1}$$

where the inequality holds due to Lemma E.5; we conclude that $\forall\delta\in(0,1/10]$, with probability at least $1-\delta$,

$$\mathfrak{m}_0(\nu,X^n)\leq\frac{4|\mathcal{S}|}{9n}+\frac{3\sqrt{|\mathcal{S}|}\log(1/\delta)}{n}$$

by substituting (E.1) into Lemma E.6. $\qquad\square$

### E.3 Upper Bounds for Binomial$(n,p)$

The following two bounds for $X\sim\mathsf{Binomial}(n,p)$ both follow from $\mathbb{E}[z^X]=(1-p+pz)^n,\forall z\in\mathbb{R}$.

**Lemma E.8.** Let $X\sim\mathsf{Binomial}(n,p)$. If $p\in(0,1]$,

$$\mathbb{E}\frac{1}{X+1}\leq\frac{1}{p(n+1)}.$$

*Proof.* This folklore (Canonne, 2020) derives from an observation that by Fubini's Theorem,

$$\mathbb{E}\frac{1}{X+1} = \int_0^1 \mathbb{E}[z^X]dz,$$

whose RHS is

$$\int_0^1 (1-p+pz)^n dz = \frac{(1-p+pz)^{n+1}}{p(n+1)}\bigg|_0^1 = \frac{1-(1-p)^{n+1}}{p(n+1)} \leq \frac{1}{p(n+1)}.$$

$\square$

**Lemma E.9.** Let $X \sim \text{Binomial}(n,p)$. For any $\eta \in (0,1)$,

$$\mathbb{P}(X \leq (1-\eta)np) \leq \exp\left(-\frac{\eta^2 np}{2}\right).$$

*Proof.* A combination of Mitzenmacher & Upfal (2017, Exercise 4.7) and the proof of Mitzenmacher & Upfal (2017, Theorem 4.5) yields the upper bound, which we provide here for completeness. For any $t < 0$, by Markov's inequality,

$$\mathbb{P}(X \leq (1-\eta)np) = \mathbb{P}\left(e^{tX} \leq e^{t(1-\eta)np}\right) \leq \frac{\mathbb{E}[e^{tX}]}{e^{t(1-\eta)np}} = \frac{(1+p(e^t-1))^n}{e^{t(1-\eta)np}}$$

$$\leq \frac{\exp(np(e^t-1))}{e^{t(1-\eta)np}} = \left(\frac{\exp(e^t-1)}{e^{t(1-\eta)}}\right)^{np} = \left(\frac{e^{-\eta}}{(1-\eta)^{1-\eta}}\right)^{np},$$

where the last inequality follows from $1+x \leq e^x, \forall x \in \mathbb{R}$ and in the last equality we set $t = \log(1-\eta)$. It remains to show

$$-\eta - (1-\eta)\log(1-\eta) \leq -\frac{\eta^2}{2}, \forall \eta \in (0,1). \tag{E.2}$$

We thereby define $f(\eta) := -\eta - (1-\eta)\log(1-\eta) + 0.5\eta^2$. A direct calculation gives

$$f'(\eta) = \log(1-\eta) + \eta, f'(0) = 0;$$
$$f''(\eta) = -\frac{1}{1-\eta} + 1 < 0, \forall \eta \in (0,1);$$

So $f$ is nonincreasing in $[0,1)$ and thus (E.2) holds. $\square$

Lemma E.9 helps us obtain a high-probability counterpart of Lemma E.8.

**Corollary E.10.** Let $X \sim \text{Binomial}(n,p)$ and $p > 0$. For any $\delta \in (0,1)$, if

$$\eta = \sqrt{\frac{2\log(1/\delta)}{np}} < 1,$$

then with probability at least $1 - \delta$,

$$\frac{1}{X+1} \leq \frac{1}{(1-\eta)np+1}.$$

*Proof.* By Lemma E.9,

$$\mathbb{P}\left(\frac{1}{X+1} > \frac{1}{(1-\eta)np+1}\right) \leq \mathbb{P}(X \leq (1-\eta)np) \leq \exp\left(-\frac{\eta^2 np}{2}\right) = \delta.$$

$\square$

## F    HARD-TO-LEARN INSTANCES FOR EXPERIMENTS

**Instance 0**    For every $s \in \mathcal{S}$, $\pi^\star(\cdot|s) := 0.5\mathsf{Uniform}(\mathcal{A}) + 0.5\mathsf{Dirac}(\mathcal{A}, s \mod A + 1)$ because any reference policy far away from both $\mathsf{Uniform}(\mathcal{A})$ and any one in $\mathsf{Dirac}(\mathcal{A})$ is sufficient to reveal the disadvantage of $\widehat{\pi}_{\mathsf{CE,ptl}}$ according to Theorem 4.3. $\rho := \mathsf{Uniform}(\mathcal{S})$ is enough to ensure about $n/s$ visitations of each input.

Interestingly, we conjecture there does not exist a worst-of-three-worlds instance that can simultaneously expose the fundamental limits of $\widehat{\pi}_{\mathsf{CE,sgl}}$, $\widehat{\pi}_{\mathsf{SEL,ptl}}$, and $\widehat{\pi}_{\mathsf{CE,ful}}$, in that the constructive proofs (in Appendix C) of Theorem 3.1, Theorem 4.1, and Theorem 5.1 since the progressively richer information are substantially different from each other. Since our learners in this section is uniformly initialized over unseen labels, any single instance covered by the Bayes prior in the lower bound arguments of a setting[14] is sufficient to numerically illustrate the corresponding difficulty of estimation (learning).

**Instance 1**    To verify the minimax optimality of $\widehat{\pi}_{\mathsf{CE,sgl}}$ with only samples avaiable, we adapt the proof of Theorem 3.1 (Appendix C.2), which is based on Assouad's hypercube reduction (Yu, 1997). In numerical simlations, any vertex of the hypercube is applicable since we have already enforced an uniform initialization of any $\widehat{\pi}$ in unseen inputs. We choose the teacher policy

$$\pi^\star(2j-1|s) = \frac{1+0.25\sqrt{SA/n}}{A}, \pi^\star(2j|s) = \frac{1-0.25\sqrt{SA/n}}{A}, \forall (s,j) \in \overline{[S]} \times \left[\frac{A}{2}\right];$$

and $\rho = \mathsf{Uniform}(\mathcal{S})$ for simplicity. The two key insights behind the design of Instance 1 are (1) $\rho$ must be nonvanishing in $\Omega(|\mathcal{S}|)$ inputs to manifest the hardness of $|\mathcal{S}| > 0$, (2) $\mathsf{TV}(\pi^\star(\cdot|s), \mathsf{Uniform}(\mathcal{A})) = \Theta(n^{-0.5})$ is crucial for Instance 1 to be hard enough for any minimax optimal learner. (If $|\mathcal{A}|$ is odd, simply let the last label $A$ to have zero mass and replace $A$ with $A-1$ here.)

**Instance 2**    To verify the minimax optimality of $\widehat{\pi}_{\mathsf{SEL,ptl}}$ with sampled odds avaiable, we adapt the proof of Theorem 4.1 (Appendix C.3), which is based on a carefully designd Bayes prior. Similarly, we can use any single instance covered by the support of the Bayes prior. We choose the teacher policy

$$\pi^\star(2j-1|s) = \frac{S}{n+1}, \pi^\star(2j|s) = 0, \pi^\star(A|s) = 1 - \frac{S}{2} \cdot \frac{A-1}{n+1}, \forall (s,j) \in \overline{[S]} \times \left[\frac{A-1}{2}\right];$$

and $\rho = \mathsf{Uniform}(\mathcal{S})$ for simplicity. (If $|\mathcal{A}|$ is even, simply let the last label $A$ to have zero mass and replace $A$ with $A-1$ here.)

**Instance 3**    To verify the minimax optimality of $\widehat{\pi}_{\mathsf{CE,ful}}$ with complete logits avaiable, we adapt the proof of Theorem 5.1 (Appendix C.4), which includes a specialized $\rho$ to slow down the convergence of $\widehat{\pi}_{\mathsf{CE,ful}}$. Specifically, $\rho = (n+1)^{-1}$ for all inputs except the last one. Theoretically, the assignment of $\pi^\star$ will not affect the convergence of $\widehat{\pi}_{\mathsf{CE,ful}}$, so we use a $\pi^\star$ same as that in Instance 3 only to ensure that $\widehat{\pi}_{\mathsf{SEL,ptl}}$ is not able to converge too fast.

## G    DISCUSSIONS: DEPENDENT SAMPLES IN REWARDLESS MDPS

Besides the popular approach of fine-tuning LLMs (Ouyang et al., 2022; Touvron et al., 2023b) that interprets instructions as inputs and the entire response as a label, there is a more granular perspective where each token is considered a label $a_i$ (See, e.g., the `logprobs` option in the OpenAI completion API[15].) and $s_{i+1}$ is simply the concatenation of $s_i$ and $a_i$, i.e., $s_{i+1} \sim \mathbb{P}(\cdot|s_i, a_i)$, where $\mathbb{P}$ is the deterministic transition kernel induced by concatenation. Our bounds for i.i.d. samples **already** subsume this plausible more involved case through *lack of reward*. The following reductions hold for any $\mathbb{P} \in \Delta(\mathcal{A}|\mathcal{S} \times \mathcal{A})$ including the aforementioned concatenation kernel.

The proof of any lower bound remains valid so long as $\mathbb{P}(\cdot|s, a) := \rho(\cdot), \forall (s, a) \in \mathcal{S} \times \mathcal{A}$ in the constructed hard-to-learn instance, making the samples i.i.d. Our upper bounds allow $\rho$ to explicitly depend on $\pi^\star$ and even $\mathbb{P}$, so the samples can be viewed as i.i.d. samples from $d_{\pi^\star}^{\mathbb{P}} \times \pi^\star$ given the *input*

---

[14]See, e.g., Appendix C.3 for a concrete Bayes prior in use.

[15]https://platform.openai.com/docs/api-reference/completions/create# logprobs

*occupancy measure* $d^{\mathbb{P}}_{\pi^\star} \in \Delta(\mathcal{S})$ is well-defined. Hence, replacing $\rho$ with $d^{\mathbb{P}}_{\pi^\star}$ validates all arguments for upper bounds. Intuitively, $d^{\mathbb{P}}_{\pi^\star}(s)$ is the probability of visiting $s$ in a trajectory induced by the transition kernel $\mathbb{P}$ and reference policy $\pi^\star$. It is well-developed in either episodic MDPs (Yin et al., 2021) or discounted MDPs (Rashidinejad et al., 2021). These seamless reductions crucially hinge on the absence of value functions and any notion of reward signal in our theoretical framework.

**Remark G.1.** Our analysis covers but is not specialized to the case where $\rho$ depends on $\pi^\star$ or vice versa. Therefore, the result remains unchanged regardless of the relation between $\rho$ and the original training set for training the teacher $\pi^\star$. (For example, $\rho$ may be the distribution of instructions selected by maintainers on the student side (Peng et al., 2023).) It will be intriguing if some further analysis can show any impact of teacher training or data quality on the students' statistical rate.

# H   DICUSSIONS ON FUNCTION APPROXIMATION

## H.1   LOG-LINEAR AND GENERAL SOFTMAX CONDITIONAL DENSITIES

We discuss potential ways and obstacles of generalizing the results above to large or even uncountable (continuous) state space. First, we extend the concept of conditional probability space $\Delta(\cdot|\cdot)$ to general spaces rigorously. In the following discussions, we assume the notation $\mathcal{Y}$ and $\mathcal{A}$ refer to finite sets for simplicity. $\Delta(\cdot)$ in this section receives any standard Borel space as input and returns the set of probability measures on it.

**Definition H.1** (Polyanskiy & Wu 2022, Definition 2.8). Given two standard Borel spaces $\mathcal{X}, \mathcal{Y}$, a conditional probability (the teacher/student we considered in this paper) $\pi : \mathcal{X} \to \mathcal{Y}$ is a bivariate function $\pi(\cdot|\cdot)$, whose first argument is a measurable subset of $\mathcal{Y}$ and the second is an element of $\mathcal{X}$, such that:

- $\forall x \in \mathcal{X}, \pi(\cdot|x)$ is a probability measure on $\mathcal{Y}$, and

- $\forall$ measurable $A \subset \mathcal{Y}, x \to \pi(A|x)$ is a measurable function on $\mathcal{X}$.

The following preliminary result (whose proof is deferred to Section H.3) may shed light on prospective approaches to the analysis of function approximation.

**Proposition H.2.** For standard Borel spaces $\mathcal{X}$ and $\mathcal{Y}$, if both $\acute{\pi}, \grave{\pi} \in \Delta(\mathcal{Y}|\mathcal{X})$ are log-linear, i.e., $\acute{\pi} = \Pi(\phi, \acute{\theta}), \grave{\pi} = \Pi(\phi, \grave{\theta})$, where

$$\Delta(\mathcal{Y}|\mathcal{X}) \ni \Pi(\phi, \theta) \propto \exp(\langle \phi, \theta \rangle), \phi : \mathcal{X} \times \mathcal{Y} \to \mathbb{R}^d;$$

and $\sup_{(x,y) \in \mathcal{X} \times \mathcal{Y}} \|\phi(x,y)\|_2 \leq M$; then for any $\nu \in \Delta(\mathcal{X})$,

$$\mathsf{TV}(\acute{\pi}, \grave{\pi}|\nu) \leq M \left\| \acute{\theta} - \grave{\theta} \right\|_2. \tag{H.1}$$

## H.2   TAKE-HOME MESSAGES AND CONJECTURES

Obviously, any analysis leveraging Proposition H.2 can potentially generalize our results, since log-linear $\pi^\star$ subsumes tabular $\pi^\star$. Technically, (H.1) mainly hinges on the (uniform) $M$-Lipschitz continuity of $\langle \phi, \theta \rangle$ with respect to $\theta$ for any $\theta \in \mathbb{R}^d$, therefore, it is also conceptually straightforward to extend Proposition H.2 to general Sofxmax $\pi^\star$, which we omit here for brevity.

Based on (H.1), we conjecture that a fine-grained analysis of the $\ell_2$-norm of $\widehat{\theta} - \theta^\star$ may be the key to **bound** $\mathsf{TV}(\widehat{\pi}, \pi^\star|\rho)$ **from above** for any $\widehat{\pi}, \pi^\star \in \Delta(\mathcal{A}|\mathcal{S})$ and $\rho \in \Delta(\mathcal{S})$. Since the tabular setting is a special case of the log-linear setting, we also conjecture that $\left\| \widehat{\theta} - \theta^\star \right\|_2 \gtrsim \sqrt{d/n}$ via Hard Labels and $\left\| \widehat{\theta} - \theta^\star \right\|_2 \gtrsim d/n$ via Partial SLs.

## H.3 PROOF OF PROPOSITION H.2

*Proof of Proposition H.2.* For any $x \in \mathcal{X}$,

$$
\begin{aligned}
\mathsf{KL}(\acute{\pi}(\cdot|x)\|\grave{\pi}(\cdot|x)) &= \sum_{y\in\mathcal{Y}}\acute{\pi}(y|x)\log\frac{\acute{\pi}(y|x)}{\grave{\pi}(y|x)} \\
&= \sum_{i\in\mathcal{Y}}\acute{\pi}(i|x)\left[\left\langle\phi(x,i),\acute{\boldsymbol{\theta}}-\grave{\boldsymbol{\theta}}\right\rangle+\log\frac{\sum_{k\in\mathcal{Y}}\exp\left(\left\langle\phi(x,k),\grave{\boldsymbol{\theta}}\right\rangle\right)}{\sum_{j\in\mathcal{Y}}\exp\left(\left\langle\phi(x,j),\acute{\boldsymbol{\theta}}\right\rangle\right)}\right] \\
&\leq \sum_{i\in\mathcal{Y}}\acute{\pi}(i|x)\left\langle\phi(x,i),\acute{\boldsymbol{\theta}}-\grave{\boldsymbol{\theta}}\right\rangle+\sum_{j\in\mathcal{Y}}\frac{\exp\left(\left\langle\phi(x,j),\grave{\boldsymbol{\theta}}\right\rangle\right)}{\sum_{k\in\mathcal{Y}}\exp\left(\left\langle\phi(x,k),\grave{\boldsymbol{\theta}}\right\rangle\right)}\left\langle\phi(x,j),\grave{\boldsymbol{\theta}}-\acute{\boldsymbol{\theta}}\right\rangle \\
&= \sum_{i\in\mathcal{Y}}(\acute{\pi}(i|x)-\grave{\pi}(i|x))\left\langle\phi(x,i),\acute{\boldsymbol{\theta}}-\grave{\boldsymbol{\theta}}\right\rangle \\
&\leq \sum_{i\in\mathcal{Y}}|\acute{\pi}(i|x)-\grave{\pi}(i|x)|\|\phi(x,i)\|_2\left\|\acute{\boldsymbol{\theta}}-\grave{\boldsymbol{\theta}}\right\|_2 \\
&\leq 2\mathsf{TV}(\acute{\pi}(\cdot|x),\grave{\pi}(\cdot|x))\cdot M\cdot\left\|\acute{\boldsymbol{\theta}}-\grave{\boldsymbol{\theta}}\right\|_2,
\end{aligned}
$$
(H.2)

where the first inequality holds due to the log-sum inequality, the second inequality is a combination of triangle inequality and Cauchy–Schwarz inequality, and the last inequality is by the boundedness of $\phi$ together with the well-known $2\mathsf{TV}=\ell_1$ relation. We plug (H.2) into Pinsker's inequality to obtain

$$
[\mathsf{TV}(\acute{\pi}(\cdot|x),\grave{\pi}(\cdot|x))]^2\leq\frac{1}{2}\mathsf{KL}(\acute{\pi}(\cdot|x)\|\grave{\pi}(\cdot|x))\leq M\mathsf{TV}(\acute{\pi}(\cdot|x),\grave{\pi}(\cdot|x))\left\|\acute{\boldsymbol{\theta}}-\grave{\boldsymbol{\theta}}\right\|_2.
$$

So $\mathsf{TV}(\acute{\pi},\grave{\pi}|\nu)=\int_{\mathcal{X}}\mathsf{TV}(\acute{\pi}(\cdot|x),\grave{\pi}(\cdot|x))\mathrm{d}\nu\leq M\left\|\acute{\boldsymbol{\theta}}-\grave{\boldsymbol{\theta}}\right\|_2.$ □

