# OpenReview forum: "Towards the Fundamental Limits of Knowledge Transfer over Finite Domains"
_ICLR.cc/2024/Conference — ICLR 2024 poster_

### Official Review · Reviewer_v1b9 · 2023-10-25

**Soundness:** 3 good
**Presentation:** 3 good
**Contribution:** 2 fair
**Rating:** 6
**Confidence:** 3

**Summary:**

The authors consider the problem of knowledge transfer between a teacher (e.g., LLMs) and a student classifier (given limited privileged information). They show the fundamental limit of the transfer under three regimes:
1. only input-label pairs are observed. The optimal rate is $\sqrt{\frac{|S||A|}{n}}$, achieved by MLE.
2. teacher probabilities of sampled labels are also available, then the optimal rate is $\frac{|S||A|}{n}$, achieved by empirical SEL loss
3. complete logits are also given, then the optimal rate is $\frac{|S|}{n}$, achieved by KL divergence minimization
Numerical simulations are also provided.

**Strengths:**

quality: high, provide thorough investigation under three regimes, including matching upper and lower bounds.
clarity: good, the setting is easy to under stand, and the structure of the paper is clear

**Weaknesses:**

The setting of "knowledge transfer" seems to be fancy. But according to the definition, the teacher distribution is considered as "ground truth" that is already given, thus the "hard labels" regime seems to fall in the common statistical learning framework. I am not sure whether tabular settings under usual statistical learning framework have been studied.

**Questions:**

1. What is the main difference between your setting and the usual statistical learning setting? For example, if I change your notation in the following way: "input" $s$ change to "feature" $x$, label $a$ change to $y$, then your setting (hard labels) seems to be completely the same as the usual setting of observing data $\\{(x_i,y_i)\\}\_{i=1}^{n}$, and the goal is to predict $y$ from $x$. It seems that there is not "knowledge transfer" (unless you call the usual machine learning "knowledge transfer from ground truth").

2. The results is fully tabular. Can you generalize your results to continuous case?

---

> ### Author Response · Authors · 2023-11-12
> **Rebuttal by Authors**
>
> Thank you for your positive feedback and let us address your questions as follows.
>
> **Q1**: I am not sure whether tabular settings under usual statistical learning framework have been studied.
>
> **A1**:
>
> - As mentioned at the end of our Section 1.1, previous works [1, 2] similar to our $\mathsf{H}\text{ard }\mathsf{L}\text{abels}$ setting consider the teacher to be also an estimator that learning from samples generated via the Bayes (ground truth) probability. [1, 2] also largely fall into the ERM (empirical risk minimization) framework. **In contrast**, motivated by modern (proprietary) language (teacher) models, we treat the teacher model to be the Bayes (ground truth) probability and our target of interest is just a kind of symmetric *closeness metric* between the student and the teacher, and thus is different from the ERM-related ones that mainly concern *generalization bounds* or the comparison between the generalization bound of the student and that of the teacher.
>
> - The term "knowledge transfer" we considered is to transfer any information from the teacher classifier to the student classifier as we mentioned in our Introduction section, any portion of $\{\pi^\star(\cdot|s_i)\}_{i=1}^n$ is considered as privileged (i.e., additional) information.
>
> - At the very beginning of Section 3 (for $\mathsf{H}\text{ard }\mathsf{L}\text{abels}$), we already emphasize that
>
>   > "This is equivalent to the standard setting for estimating the conditional density $\pi^\star(\cdot|\cdot)$"
>
> **Q2**: Can you generalize your results to continuous case?
>
> **A2**: In the case of log-linear models $\pi_{{\theta}}(\cdot|\cdot) \propto \exp(\langle {\theta}, {\phi}(\cdot, \cdot))$, we do have some preliminary results that the conditional total variation between two models can be upper bounded by the $\ell_2$-norm between the model parameters given uniformly bounded feature mapping ${\phi}$. We have added these discussions and related conjectures (for log-linear or general softmax $\pi^\star$) on function approximation when the state space is allowed to be any standard Borel space in Appendix H (highlighted in blue) and leave the complete characterization as future work.
>
> References:
>
> [1] Aditya Krishna Menon, Ankit Singh Rawat, Sashank J Reddi, Seungyeon Kim, and Sanjiv Kumar.
> Why distillation helps: a statistical perspective. arXiv preprint arXiv:2005.10419, 2020.
>
> [2] Tri Dao, Govinda M Kamath, Vasilis Syrgkanis, and Lester Mackey. Knowledge distillation as
> semiparametric inference. arXiv preprint arXiv:2104.09732, 2021.

---

### Official Review · Reviewer_2SeV · 2023-10-28

**Soundness:** 4 excellent
**Presentation:** 4 excellent
**Contribution:** 2 fair
**Rating:** 6
**Confidence:** 3

**Summary:**

This paper attempts to create a mathematical model for a simplified version of "knowledge transfer" from a teacher to a student within the context of multi-class learning. Notably, the authors make two key assumptions: that both 1) $\mathcal{S}$ (representing the space of all possible queries a student can pose to a teacher) and 2) $\mathcal{A}$ (denoting the space of all possible answers provided by the teacher) are finite.

In mathematical terms, for each query $s\in\mathcal{S}$, let $\pi^*(\cdot\vert s)$ represent the (optimal) teacher policy, or the conditional distribution of labels for the query $s$. The student's goal is to learn $\hat{\pi}$, a finite distribution with the same complexity as $\pi^*$, using only $n$ i.i.d. queries (and answers) from the teacher, represented by the set $\\{(a_i,s_i)\\}$ for $i\in[n]$.

In this context, the authors aim to establish minimax bounds for $\mathbb{E}_{s\sim\rho}\left[\mathsf{TV}\left(\hat{\pi},\pi^*\vert\rho\right)\right]$, which measures the "$\rho$-expected total variation distance" between the teacher and student's policies. Here, $\rho$ signifies the distribution that generates queries across $\mathcal{S}$. The paper successfully derives minimax rates for this problem under three different scenarios: i) when only hard labels are provided, ii) in cases where, in addition to labels, the probability of the label from $\pi^*$ is also given, and iii) when all class probabilities are given for each query.

The paper is well-written, and it meticulously references pertinent prior research (as far as I can tell). The mathematical derivations appear sound, with no conspicuous errors. Moreover, there are several experiments, though I haven't examined them closely. Overall, this is a commendable theoretical contribution. However, I believe that the assumption of finiteness for both the "query" and "answer" spaces oversimplifies the problem, leading to bounds that might not be as intriguing as one might hope (please refer to the weaknesses section for further clarification).

My vote is a weak accept.

**Strengths:**

- The paper is very well-written. In particular, the "Prior Works" section stands out as highly informative.

- The problem formulation and motivations, especially the connections to Large Language Models (LLMs) and other foundational models, are immensely intriguing.

- The theoretical framework developed in this work appears robust and sufficiently rigorous for a theory paper at ICLR. I scrutinized some of the proofs and did not find any significant errors. Furthermore, the other results seem mathematically sound.

- The problem is analyzed under three distinct regimes, with the gradual introduction of additional side information to the original dataset. This approach allows the reader to observe, in a step-by-step manner, the mathematical impact of incorporating each layer of side information.

- The paper also includes some experimental validations, although I did not review them in detail.

**Weaknesses:**

I suppose that the assumptions regarding the finiteness of both $\mathcal{S}$ and $\mathcal{A}$ may have oversimplified the problem. Here is my interpretation of Theorems 3.1 and 3.2 (please correct me if I am mistaken):

- In essence, the paper aims to "learn" a distribution of dimension ($|\mathcal{S}|\times|\mathcal{A}|$), denoted as $\rho\times\pi^*$. By "learn," I mean the process of minimizing a specific expected total variation (TV) distance, as mentioned earlier. Let me introduce the notation $F = |\mathcal{S}|\times|\mathcal{A|}$. It is already established that having approximately $\tilde{O}(F)$ (where $\tilde{O}$ hides polylogarithmic factors) samples, denoted as $n$, is sufficient to capture almost all the probability mass within $\mathcal{S} \times \mathcal{A}$ according to the distributions $\rho$ and $\pi^*$. This can be directly derived from the coupon collector theorem. Consequently, in the worst-case scenario where both $\rho$ and $\pi^*$ are not sparse, we typically have, on average, $\tilde{O}(n/F)$ samples for each query-answer pair $(a, s)$. Consequently, the overall distribution $\rho \times \pi^*$, as well as any of its derivatives (such as the aforementioned "expected TV distance," which is the primary focus of this paper), can be approximated with an error of at most $\tilde{O}(\sqrt{F/n})$, as a direct consequence of general inequalities like Hoeffding. In my opinion, this summarizes most of Theorem 3.1 and 3.2. However, the rigorious derivation of final formulations, which has been done in this paper, is still a nice addition to MLT community.

- The bounds pertaining to other scenarios discussed in the paper can also be derived using mathematical methods similar to the ones mentioned above.

Nonetheless, I commend the authors for their precise determination of minimax rates and their detailed analysis of individual cases. It would, however, be more intriguing if the paper explored more intricate scenarios for $\mathcal{S}$ and $\mathcal{A}$. For example, one would naturally expect other (and more interesting) notions of complexity (such as VC-dimension, etc.) of distribution families $\rho$ and $\pi^*$ to show up in bounds, instead of the mere cardinality of query-answer spaces.

**Questions:**

Pleae see "Weaknesses" section.

---

> ### Author Response · Authors · 2023-11-12
> **Rebuttal by Authors**
>
> Thank you for your detailed feedback and we are happy to address your questions as follows.
>
> **Q1**: The reviewer's interpretation of Theorem 3.1, Theorem 3.2, and other results.
>
> **A1**: Let us respectfully emphasize some key points to accurately deliver the messages as follows.
>
> - The learner (student model) does not learn $\rho\times\pi^\star$, and only aim to learn $\pi^\star(\cdot|\cdot)$, which is a **discrete** and **conditional** density. $\rho\times\pi^\star$ is purely a probability distribution indexed by $(s, a)$ pairs, and learning a good $\widehat{\rho\times\pi^\star}$ **can not** imply learning a good $\widehat{\pi^\star}$.
>
> - Thus, the target of interest is the conditional total variation $\mathsf{TV}(\hat\pi, \pi^\star|\rho)$, while as we mentioned in the paper, an uniform $\rho$ is indeed good to provide some intuitions about the worst-case upper bound **in expectation** in Theorem 3.2.
>
> - We respectfully disagree with the reviewer on our contributions to understanding the method efficiency of **the later two** settings. When certain portion of $\{\pi^\star(\cdot|s_i)\}_{i=1}^n$ is provided to the student, vanilla concentration bounds concerning only samples (such as Hoeffding's inequality, the Angluin-Valiant bound, Bernstein's inequality, Bennett's inequality, or McDiarmid's theorem) are **no longer applicable**, so our novel technical roadmap is instead inspired by *the Missing Mass Analysis* [1].
>
> - We also respectfully disagree with the reviewer on our contributions to understanding the problem complexity of **each** setting. The three information-theoretic lower bounds **can not** be easily derived from standard concentration bounds. For the first case, the constructive proof of Theorem 3.1 is derived from Assouad's hypercube reduction. For the later two cases, Theorem 4.1 and Theorem 5.1 **can not** even be derived via standard reductions like Le Cam's, Assound's, or Fano's [2], because these three standard methods can only tackle the case with no ground truth leakage, i.e., they can only deal with estimators of $\pi^\star$  that learn from purely samples from $\rho\times\pi^\star$. In contrast, our proofs of Theorem 4.1 and Theorem 5.1 are based on two different and explicit constructions of Bayes priors over $\rho\times\pi^\star$ and the reduction from bounding minimax risk to bounding worse-case Bayes risk from below [3]. Thus, all the three constructive proofs for Theorem 3.1, Theorem 4.1, and Theorem 5.1 , are not derived from standard concentration bounds, and are **different** from each other, even from an intuitive perspective.
>
> - Also, we want to mention an important contribution related to **instance-dependent**/benign-case analysis in Theorem 3.2: the complexity measure $\xi(\pi^\star)$, which provably explains the $O(1/n)$ rate of $\mathsf{CE_{sgl}}$ rather than $O(1/\sqrt{n})$ **in expectation** when $\xi(\pi^\star)\asymp n^{-1}\text{poly}(|\mathcal S|, |\mathcal A|)$.
>
> - Another contribution is that the while the worst-case **high-probability** upper bound in Theorem 3.2 has $\text{poly}\log(|\mathcal S|, |\mathcal A|)$ factors, the worst-case **upper bound** in expectation in Theorem 3.2 **does not**.
>
> **Q2**: More intricate scenarios for $\mathcal S$ and $\mathcal A$?
>
> **A2**: We answer this question in two fold. First, the rationale behind considering **discrete** $\rho$ and $\pi^\star$ is that modern language models are often classifiers of **tokens**, with tokens or a collection of tokens as input. Second, we have added some discussions and conjectures (for log-linear or general softmax $\pi^\star$) on function approximation in the face of large state space in Appendix H (highlighted in blue), which builds a quantitative relation between the $\ell_2$-distance of model parameters and the conditional total variation between models, and leave the complete characterization as future work.
>
> References:
>
> [1] David McAllester and Luis Ortiz. Concentration inequalities for the missing mass and for histogram
> rule error. Journal of Machine Learning Research, 4(Oct):895–911, 2003.
>
> [2] Bin Yu. Assouad, fano, and le cam. In Festschrift for Lucien Le Cam: research papers in probability
> and statistics, pp. 423–435. Springer, 1997.
>
> [3] Yury Polyanskiy and Yihong Wu. Information theory: From coding to learning. Book draft, 2022.

---

> > ### Comment · Reviewer_2SeV · 2023-11-19
> >
> > I would like to thank the authors for addressing my comments. They have addressed the weaknesses I raised at three nearly distinct points in their response:
> >
> > - The first concerns the distinction between learning $\rho\times\pi^*$ and $\pi^*\left(\cdot\vert\cdot\right)$ (although the latter needs to be learned in a way to minimize some given expected loss with respect to $\rho$). The authors rightly assert that these are distinct problems. I did not intend to imply otherwise. However, it appears that they are closely related "in essence." In the **tabular** setting you've presented, both are tables of size $\vert\mathcal{A}\vert\times\vert\mathcal{S}\vert$, and they are tightly related through simple mathematical relations. I remain unconvinced that learning one is fundamentally different from learning the other. In other words, I can revisit my initial comments and substitute all the statements related to $\rho\times\pi^*$ with analogous statements about $\pi^*\left(\cdot\vert\cdot\right)$, and they still resonate with me (at least). This issue has been raised by another reviewer as well, that assuming finite $\mathcal{A}$ and $\mathcal{S}$ (tabular setting) might be too basic.
> >
> > - The second point mentioned by the authors revolves around the authors' idea that including what they call 'privileged information' fundamentally changes the problem, making traditional tools (such as traditional concentration bounds, etc.) less effective. And therefore, the technical contributions around this matter might be significant.
> >
> > - Similarly, the third point suggests that the tools used to come up with minimax bounds in this paper are quite different from the usual methods like Le Cam's, Assound's, or Fano's techniques. Again, showcasing the high level of mathematical sophistication in this work.
> >
> > I'm not entirely sure about the second and third points right now since I haven't delved deeply into the paper. However, it's worth mentioning that these points don't seem to be highlighted enough in the paper, possibly getting lost in the midst of all the technical details. So, I kindly recommend that the authors consider a major rewrite to better showcase these contributions.
> >
> > Additionally, I suggest that the authors take into consideration the feedback from Reviewer v1b9 regarding the definition and use of 'knowledge transfer' in this work. It seems that the primary focus of the authors' contribution lies more in the realm of "learning" with privileged information rather than what is commonly understood as "knowledge transfer" in the machine learning community today. In your study, the teacher is treated as the ground truth, and the student's ability to emulate the 'true' optimal classifier solely by accessing the teacher is somewhat diminished.
> >
> > I maintain my vote, which leans towards acceptance

---

> ### Author Response · Authors · 2023-11-22
> **Thank you for your suggestions!**
>
> We are grateful for your attentive replies and would like to further address the points you mentioned as follows.
>
> - In the latest revision, we highlighted brief discussions on the technical novelty of our work in the original text in blue; namely, the corresponding paragraph in Section 1.1 and the corresponding sentence in Section 3.1. Due to space limit, we may also consider further detailing the technical arguments in Appendix in subsequent revisions.
>
> - To inspire future work on function approximation, we have highlighted our newly added quantitative inequality relationship between the $\ell_2$ distance of two model parameters (assuming both models are log-linear) and the conditional total variation of the corresponding two models; and have proposed relevant conjectures on the lower bound of the $\ell_2$ distance of two model parameters.
>
> - Regarding the term "knowledge transfer", we would like to mention that the three cases we analyzed are idealized versions of "knowledge distillation" in deep learning practice [3]. Recently, there has been a trend of using (prompt, response) pairs queried from (Chat)GPT(-4) to train open-source language models, for example, see [1, 2]. These "black-box" knowledge transfer (distillation) paradigms can be essentially idealized to the first case we considered.
>
> Thank you so much for your nice and scrupulous review.
>
> References
>
> [1] Zheng, L., Chiang, W.-L., Sheng, Y., Zhuang, S., Wu, Z., Zhuang, Y., Lin, Z., Li, Z., Li,
> D., Xing, E. P., Zhang, H., Gonzalez, J. E. and Stoica, I. (2023). Judging llm-as-a-judge
> with mt-bench and chatbot arena. https://github.com/lm-sys/FastChat/
>
> [2] Wang, G., Cheng, S., Yu, Q. and Liu, C. (2023a). OpenChat: Advancing Open-source Language
> Models with Imperfect Data. https://github.com/imoneoi/openchat
>
> [3] Hinton, G., Vinyals, O. and Dean, J. (2015). Distilling the knowledge in a neural network.
> arXiv preprint arXiv:1503.02531 .

---

### Official Review · Reviewer_CQ5M · 2023-11-01

**Soundness:** 4 excellent
**Presentation:** 4 excellent
**Contribution:** 3 good
**Rating:** 8
**Confidence:** 4

**Summary:**

This paper considers the problem of knowledge transfer through empirical samples from a teacher to a student. They characterize sample complexity for three different settings. The first setting is hard labels, where the student knows (input, label) samples from the teacher, the second setting is when the student knows (input, label, prob(label|input)) samples, and in the third setting, the student knows (input, label, prob(.|input)). The paper characterizes the lower bound and matching upper bounds in each of the settings, showing the statistical roles of extra information present to the student.

**Strengths:**

- Neat results and message

**Weaknesses:**

- The tabular setting is quite basic which does not capture many practical settings when the state space is large.

**Questions:**

* How can these guarantees generalize to more practical settings where the state space is large? (speculation is fine)

---

> ### Author Response · Authors · 2023-11-12
> **Rebuttal by Authors**
>
> Thank you for your appreciation and we have added some discussions and conjectures (for log-linear or general Softmax $\pi^\star$) on function approximation in the face of large $\mathcal S$ in Appendix H (highlighted in blue), which builds a quantitative relation between the $\ell_2$-distance of model parameters and the conditional total variation between models.

---

### Official Review · Reviewer_ADJF · 2023-11-01

**Soundness:** 2 fair
**Presentation:** 2 fair
**Contribution:** 2 fair
**Rating:** 6
**Confidence:** 2

**Summary:**

The authors provide minimax rates for transfer learning, expressed as the total variation between between a learner and a reference policy or a ground truth distribution, for different levels of knowledge shared with the learner and losses such as cross entropy and squared loss. They also present simulations to demonstrate the performance of their results in the various transfer settings.

**Strengths:**

* Their work explores various ways of sharing knowledge and allows for flexibility on how teachers can assist learners.
* Their theoretical results show that learners might not require too much data. Since their results are for commonly used losses in practice, this can be helpful in engineering other ways of knowledge sharing.

**Weaknesses:**

For someone who is not familiar with the literature in this field, many notations lack proper definitions or sufficient rigor. For example, there is no clear definition provided for "privileged information," and it would be beneficial for completeness to include such definitions. The technical aspects of their three settings are not distinctly emphasized. Many variables, such as $CE_{sgl}$ or $CE_{ful}$, are referenced without prior definitions. Furthermore, $\Delta(\mathcal{A} \vert \mathcal{S})$ is mentioned without an initial definition, and it's only later clarified that $\Delta(\mathcal{A})$ represents a simplex.

It took multiple reads to understand the impact of their main contribution. This paper might need several revisions because as it stands, it is not easily understandable.

**Questions:**

It would be helpful to have more specific explanation for the settings of transfer using Partially Soft Labels and Soft Labels in sections 3 and 4. It looks like $\mathcal{D}$ is already generated using $(\rho \times \pi^\star)^n$. It is not clear what additional information is provided with $\pi^\star(a_i\vert s_i)$ in the case of partially soft labels.

---

> ### Author Response · Authors · 2023-11-12
> **Rebuttal by Authors**
>
> Thank you for your valuable feedback. We would like to address your concerns as follows.
>
> **Q1**: What is the definition of "privileged information"?
>
> **A1**: Our formulations, which can indeed be viewed as concrete instantiations of the *learning using privileged information* (LUPI) framework [1], are already illustrated via relevant LLM examples in the third paragraph. To be more precise, **the privileged information we considered** for one sample pair $(s_i, a_i)$ is defined as certain portion of $\pi^\star(\cdot|s_i)$. We apologize for the vague introduction of this concept in the third paragraph of the Introduction and have incorporated your suggestions (highlighted in blue at the end of Introduction) and polished related descriptions in the current revision.
>
> **Q2**: Notation referenced before definition.
>
> **A2**: For all the four losses, we have equip them with hyperlinks (highlighted in red) pointing to corresponding definitions in the current revision, especially in Table 1. We also put the formal definition of $\Delta(\cdot|\cdot)$ and $\Delta(\cdot)$ back to the main text (highlighted in blue), which were placed in the Appendix before Appendix A due to limited space. Thank you for helping us improve the writing.
>
> **Q3**: What additional information is provided with $\pi^\star(a_i|s_i)$  in the case of partially soft labels?
>
> **A3**:  $\pi^\star(a_i|s_i)$ itself is exactly the additional information in Section 4. A plainer explanation is that one student $\hat\pi$ may want to learn $\pi^\star(\cdot|\cdot)$, and in Section 3 she only use $\mathcal{D}$, while in Section 4 she can utilize $\mathcal{D}\cup\{\pi^\star(a_i|s_i)\}_{i=1}^n$. As we mentioned in the paper, $\pi^\star(a_i|s_i)$ is the **probability** of choosing action $a_i$ conditioned on the state $s_i$ and $\pi^\star(\cdot|s_i)$ is the probability distribution over the entire action space $\mathcal A$ given the state $s_i$, where $(s_i, a_i)$ is one sample pair in $\mathcal D$. For example, suppose $\mathcal S = \{0, 1\}$, $\mathcal{A} = \{0, 1\}$, $\pi^\star(\cdot|0) = [0.3, 0.7]$, and $(s_1, a_1)$ happens to be $(0, 1)$, then $\pi^\star(a_1|s_1) = \pi^\star(1|0) = 0.7$ is the additional information (or privileged information) provided to the student together with the sample pair $(s_1, a_1)$ in the case of $\mathsf{P}\text{artial }\mathsf{SL}\text{s}$.
>
> [1] Vladimir Vapnik, Rauf Izmailov, et al. Learning using privileged information: similarity control and
> knowledge transfer. J. Mach. Learn. Res., 16(1):2023–2049, 2015.

---

> > ### Comment · Reviewer_ADJF · 2023-11-20
> >
> > I am satisfied with the authors' responses to some of the weaknesses that I mentioned and I am increasing the score to 6.

---

> > > ### Author Response · Authors · 2023-11-22
> > > **Thank you!**
> > >
> > > We are glad that our response has effectively addressed your concerns and suggestions.

---

### Meta-Review · Area_Chair_b4VU · 2023-12-24

**Metareview:**

This paper attempts to create a mathematical model for a simplified version of "knowledge transfer" from a teacher to a student within the context of multi-class learning under the assumption that both the input and output domains are finite. The paper successfully derives minimax rates for this problem under three different scenarios: i) when only hard labels are provided, ii) in cases where, in addition to labels, the probability of the label is also given, and iii) when all class probabilities are given for each query.

While this paper only studies an idealized model, all reviewers agreed that the contribution in this paper is an important step towards an important problem. The AC agrees and recommends acceptance.

**Justification For Why Not Higher Score:**

This paper only studies the finite-domain and bound scales with the domain size. Therefore, there is still a gap between this theoretical analysis and practice.

**Justification For Why Not Lower Score:**

The minimax bounds for this new setting are novel.

---

### Decision · Program_Chairs · 2024-01-16

Accept (poster)